# Light rain exacerbates extreme humid heat

Zhanjie Zhang [1], Yong Wang [1] ✉, Guang J. Zhang [2] ✉, Cheng Xing[3], Wenwen Xia[1,4] & Mengmiao Yang[5,6]

Humid heat waves pose significant risks to human health and the ecosystem. Intuitively, rainfall often alleviates extreme humid heat. However, here we show that light rain often accompanies extreme humid heat, exacerbating its frequency and intensity, especially over arid and semi-arid regions compared to no rain and moderate-to-heavy rain cases. This is because light rain does not dramatically reduce solar radiation but increases near-surface humidity through enhanced surface evaporation. The water replenishment from light rain as well as a shallower planetary boundary layer is crucial for consecutive extremes where there are commonly sporadic drizzle days amidst several rain-free days. These extremes last longer than rain-free extremes. Current global climate models (GCMs) overestimate light rain. After reducing this bias in a GCM, underestimations of humid heat waves in energy-limited regions and overestimations in water-limited regions are largely alleviated. These findings underscore the underappreciated impact of light rain on extreme humid heat.

Extreme heat events have a profound impact on human health, increasing morbidity and mortality and negatively affecting mental health[1–3]. Furthermore, extreme heat poses a significant threat to infrastructure, agriculture, and ecosystems, leading to serious economic, environmental, and societal harms[4–6]. When high temperature occurs in combination with high humidity (referred to as humid heat waves), the heat stress of these events on human health is further enhanced, resulting in increased mortality[7–10].

Many heat stress indices have been developed to synthesize the effects of temperature and humidity[11–13]. Among these, the wet bulb globe temperature (WBGT) is the most commonly used one as the international standard for estimating heat stress, as it encompasses the effects of temperature, humidity, radiation, and wind on heat stress[14–16]. In recent decades, the incidence and intensity of extreme humid heat have continued to rise[17]. Under global warming, the intensity and frequency of future humid heat waves will further increase[18,19]. Compared to extreme dry heat events, humid heat waves will be more frequent, stronger, and last longer in the future[20]. Therefore, humid heat waves will become an even greater threat as climate warms, with substantial impacts on human health, global labor, and the economy[21–23].

When humid heat waves occur, the soil is wetter than normal. Accordingly, the planetary boundary layer is shallower[24], and the accumulation of water vapor near the surface increases humidity. The occurrence of light rain affects water vapor and air temperature near the surface through the modulation of evaporation, radiation, and other processes[25,26]. How important is light rain to humid heat waves? Future projections of humid heat waves rely on global climate models (GCMs). However, current GCMs commonly suffer from excessive occurrence of light rain[27,28]. How does this model bias impact the simulated humid heat waves? What are the implications of this bias for future projections of humid heat waves in different regions?

Here, using observations and reanalysis, we show that the frequency of humid heat extremes is significantly exacerbated by light rain on and before the days of humid heat waves, and humid heat waves can be intensified by light rain mainly in arid and semi-arid regions. During consecutive humid heat waves, the uneven temporal distribution of light rain occurrence can greatly prolong their durations by sustaining surface evaporation. We further examine light rain and humid heat waves in a GCM, which has a common bias of "too much light rain and too little heavy rain" in the Coupled Model

[1]Ministry of Education Key Laboratory for Earth System Modeling and Department of Earth System Science, Tsinghua University, Beijing, China. [2]Scripps Institution of Oceanography, La Jolla, CA, USA. [3]National Key Laboratory of Microwave Imaging Technology, Aerospace Information Research Institute, Chinese Academy of Sciences, Beijing, China. [4]State Key Laboratory of Numerical Modelling for Atmospheric Sciences and Geophysical Fluid Dynamics, Institute of Atmospheric Physics, Chinese Academy of Sciences, Beijing, China. [5]Key Laboratory for Humid Subtropical Eco-Geographical Processes of the Ministry of Education, Fujian Normal University, Fuzhou, China. [6]School of Geographical Sciences, Fujian Normal University, Fuzhou, China. ✉e-mail: yongw@mail.tsinghua.edu.cn; gzhang@ucsd.edu

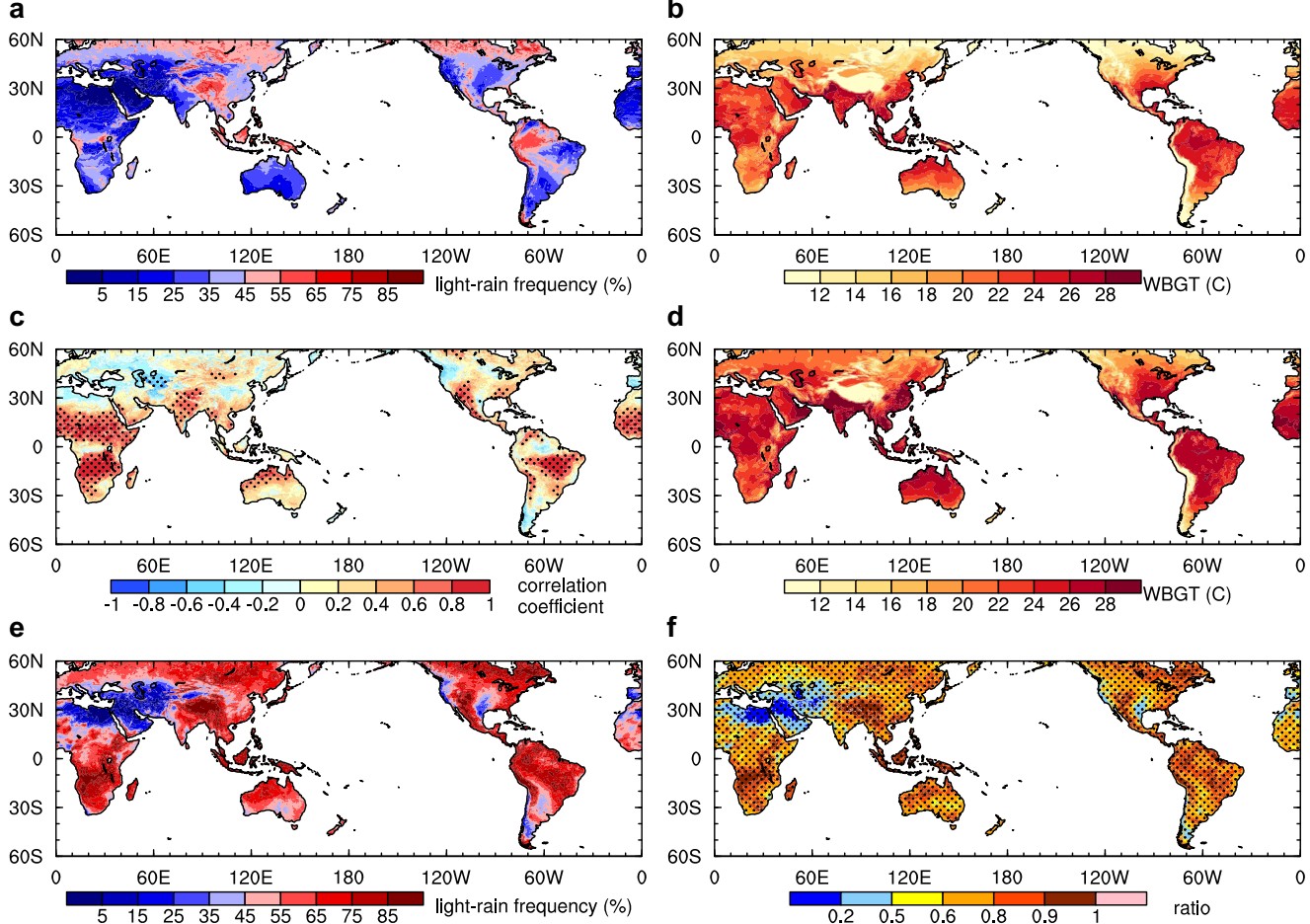

**Fig. 1 | Light rain, wet bulb globe temperature (WBGT) and their relationship during the hottest four months. a, b, c** Global distributions of light rain probability (**a**), WBGT intensity (**b**), and temporal correlation coefficients between the two at the monthly scale over 2001–2020 (**c**). **d** Global distribution of the intensity of WBGT exceeding the 95th percentile. **e** Probability of extreme humid heat days and the day before with light rain. **f** Ratios of the standard deviation of the annual number of light rain days on humid heat wave days and the day before to that of the annual number of humid heat wave days, and the positively correlated areas in time are stippled. Areas exceeding the 95% confidence level of the *t* test in (**c**) are stippled.

Intercomparison Project Phase 5 and 6 models. It is shown that excessive light rain in the GCM has distinct impacts on extreme humid heat across evaporation regimes.

## Results

### Light rain exacerbates both the frequency and intensity of humid heat waves

We analyze the distribution of the occurrence frequency of light rain (0.1–10 mm d$^{-1}$)[29] at the daily timescale and its link with heat stress index WBGT (see "Methods"). During the four hottest months of the year, defined as the four consecutive months with the highest average 2-m air temperature, the frequency of light rain is relatively high in regions near the equator and in Southeast Asia (Fig. 1a), where WBGT intensities (defined as the average of WBGT over the 4 hottest months) are high as well (Fig. 1b). There are exceptions over arid and semi-arid regions with high WBGT but low light rain frequency, which correspond to high air temperature, strong surface downward solar radiation and intense irrigation[30,31]. In the tropics and subtropics, there is a significant temporal correlation between the occurrence frequency of light rain and WBGT intensity, with correlation coefficients larger than 0.8 (Fig. 1c). Compared with the light rain frequency, the frequency of moderate to heavy rain with daily rainfall intensity greater than 20 mm d$^{-1}$[26,27] is much smaller showing maxima smaller than 20% (Supplementary Fig. 1a)[32], and its correlation with WBGT is not significant (Supplementary Fig. 1b), indicating that WBGT is not regulated by the

frequency of moderate to heavy rainfall. In this regard, the following analyses focus on the comparison between light-rain and no-rain cases.

Note that the temporal correlation between total precipitation and WBGT is insignificant or even negative, especially over monsoon regions (Supplementary Fig. 1c). This indicates that the correlation between light rain and humid heat is not a result of temporal variation of monsoon dynamics or other large-scale processes. The relationship between light rain frequency and WBGT intensity is primarily determined by the connection of light rain to natural wet bulb temperature (Supplementary Fig. 2) since natural wet bulb temperature is the dominant factor in determining the WBGT intensity. Light rain primarily affects surface evaporation and radiation processes. In areas with high frequencies of light rain, there is strong surface evaporation, resulting in more near-surface water vapor, while downward solar radiation is reduced only slightly (Supplementary Figs. 3 and 4). The distributions of the significant positive correlation between surface evaporation/near-surface water vapor and the frequency of light rain resemble that of the positive correlation between WBGT/natural wet bulb temperature and light rain frequency, suggesting that the impact of light rain is mainly through surface evaporation.

The strong correlation between light rain frequency and WBGT intensity during the warm season implies that light rain should also be correlated with humid heat extremes (defined as those days with WBGT intensities exceeding the 95th percentile over the reference period of 2001–2020). The distribution of WBGT intensity during

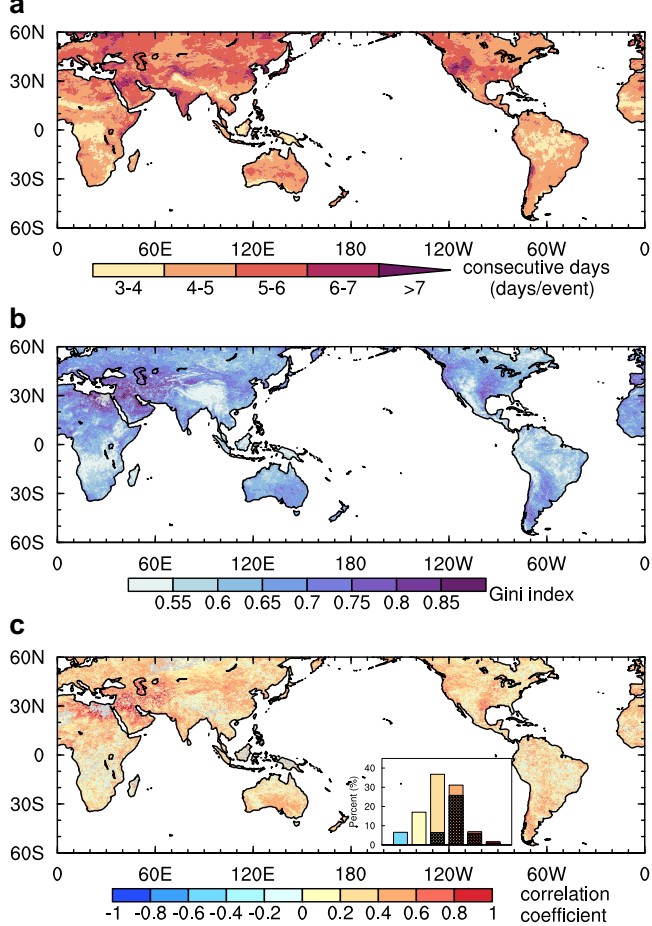

**Fig. 2 | Consecutive event duration and uneven light rain. a** Average duration of consecutive humid heat waves. **b** Gini index of light rain occurrence distribution during and the day before consecutive events. **c** Correlation coefficients between the average number of days of consecutive events and the corresponding Gini index, the inset bar chart shows the percentage of the coefficients in the intervals of ≤ 0, 0–0.2, 0.2–0.4, 0.4–0.6, 0.6–0.8 and 0.8–1.0, exceeding the 95% confidence level of the $t$ test are shaded.

with light rain is greater than that on non-rainy days (Supplementary Fig. 6a).

To screen out possible influences of interannual variabilities of large-scale dynamical processes on both light rain and humid heat, we perform a composite analysis of humid heat events within a year. Taking 2001 as an example (the results for other years are similar, figure not shown), Supplementary Fig. 6b shows the probability distribution function of averaged WBGT for both light rain and non-rainy cases. There is a higher percentage of stronger WBGT when there is light rain. The large-scale dynamical background during humid heatwaves with light rain and without rain are almost identical, characterized by the similar large-scale meteorological fields (i.e., surface pressure, 500 hPa geopotential height, and 10-m wind speed) between the two groups (Supplementary Fig. 7), which is divided by whether the selected grid cell on humid heat waves has light rain or without rain. These two figures, along with Supplementary Fig. 1c, support the conclusion that light rain exacerbates humid heat rather than that they co-vary with the variation of large-scale dynamical processes.

## Unevenness of light rain occurrence extends consecutive humid heat waves

Given that more light rain days in the warm season are associated with more humid heat waves, some of these humid heat waves are a result of consecutive humid heat waves prolonged by the incidence of light rain (Supplementary Fig. 8). We classify the events that last for three or more consecutive days with daily WBGT exceeding the 95th percentile as consecutive humid heat waves[20,33]. The distribution features of light rain in consecutive events (including the day before the event begins) and their impact on the duration and intensity of these events are analyzed. The average duration of humid heat waves in low latitudes is shorter, about 3 to 5 days, while in mid-to-high latitudes, the duration is longer, reaching six days or longer (Fig. 2a). For consecutive events with light rain, light rain prolongs humid heat more efficiently than enhancing the intensity (Supplementary Fig. 9).

To quantify the temporal distribution of light rain occurrence during consecutive events, we use the Gini index[34]. The smaller the Gini index, the more evenly distributed the rain occurrence in time (see "Methods"). We calculate the Gini index of rainfall during and a day before consecutive events (Supplementary Fig. 10a) and find it to be almost identical to the Gini index of events with only light rain occurrence (Fig. 2b). This confirms that light rain is the dominant type of precipitation in consecutive events, and the Gini index is determined by the uneven distribution of light rain occurrence[34]. In areas near the equator and the west coast of Mexico, the frequency of light rain exceeds 90% (Fig. 1e), and the Gini index is small (Fig. 2b), showing a more even distribution of light rain occurrence during consecutive events. The probability of light rain occurring during and on the day before consecutive events is over 50% in most areas, and there are typically one to three days of light rain during these events (Supplementary Fig. 10b, c). Overall, regions with more uneven distributions of light rain have longer durations of consecutive events (Fig. 2a, b). The global distribution of temporal correlation coefficients between the two shows that over one-third of the regions have coefficients exceeding 0.4, most of them statistically significant at the 95% level (Fig. 2c). This demonstrates a significant positive correlation between the uneven distribution of light rain occurrence and the duration of the events. Although the temporal correlation between the Gini index and the intensity of consecutive events is generally weaker than that with the duration (Supplementary Fig. 10d), nearly 80% of the regions show a positive correlation, with coefficients that exceed 0.4 passing the significance test. This suggests that the unevenness of light rain occurrence can affect not only the duration of consecutive humid heat waves but also their intensity. Given that more than 50% of humid heat waves are accompanied by light rain (Fig. 1e), this may be incorrectly interpreted as consecutive events having more frequent or nearly

humid heat waves is similar to that of the warm-season average, but with stronger magnitudes (Fig. 1d). The probability of light rain occurring on any given day during the extremely humid heat events and the day before such events shows that more than 50% of humid heat waves are accompanied by light rain, except over dry lands such as the north side of the Sahara Desert and the Arabian Peninsula (Fig. 1e). The rest rain-free days can still be influenced by light rain of a couple of days ago (see below).

To investigate the extent to which the frequency of humid heat waves can be augmented by an increase in light rain days, their inter-annual variabilities are analyzed. The inter-annual variation of the light rain days during the humid heat days and the day before is positively correlated with that of the number of extreme humid heat days (Fig. 1f). They both demonstrate larger inter-annual variations in tropical and subtropical regions (around 9–18 days, Supplementary Fig. 5). The ratios of the standard deviation of light rain days during the extreme humid heat days and the day before to that of humid heat waves exceed 0.5 in most regions and approach 1 in the Amazon, southern Africa, western Europe, and southern North America. These ratios indicate that 50–100% of the number of extreme humid heat days increases is contributed by the increase of light rain days. In addition, in dry areas such as northern Africa, the Arabian Peninsula, and the western United States, the intensity of extreme humid heat

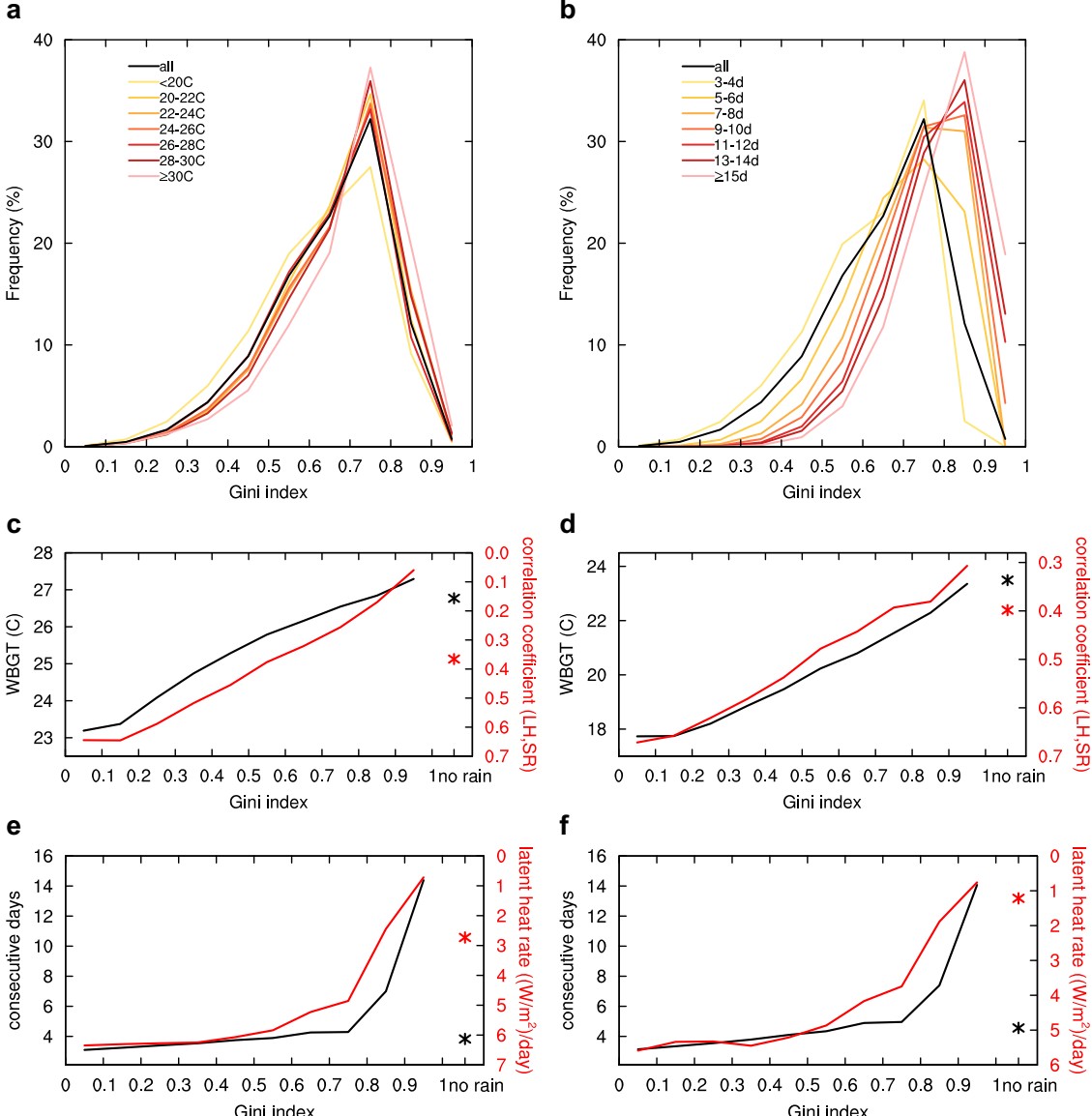

**Fig. 3 | The frequency distribution of the Gini index, and intensity and duration of consecutive events as functions of Gini indices. a, b** The frequency distribution of the Gini index for all consecutive humid heat events with light rain (black line), and separating it into those for different intensities (**a**) and durations (**b**) of consecutive humid heat events. **c, d** Wet bulb globe temperature (WBGT) intensity (black line) and correlation coefficients between surface latent heat flux (LH) and surface downwelling shortwave radiation (SR) (red line) as functions of Gini indices in the tropics (20°S-20°N) (**c**) and mid-latitudes (20°N-60°N) (**d**). **e, f** Days of consecutive events (black line) and rate of latent heat flux change (red line) as functions of Gini indices in the tropics (20°S-20°N) (**e**) and mid-latitudes (20°N-60°N) (**f**). Values for non-rainy days in **c**–**f** are plotted with asterisks.

continuous light rain. However, by employing the Gini index of precipitation (a combination of the number of rainy days and their distribution), it is found that a more uneven distribution of light rain within consecutive humid heatwaves will actually make the events stronger and last longer. The uneven temporal distribution of light rain in consecutive humid heatwaves further confirms that light rain and humid heat are not regulated by the same common factor, such as large-scale monsoon systems.

All consecutive events are grouped by their intensities and durations, though which the occurrence frequency of cases with different Gini indices in light rain events are presented (Fig. 3a, b). The frequency distribution of the Gini index for all consecutive events with light rain shows a peak at Gini index values of 0.7–0.8, with low frequencies in both tails and high frequencies in between. The distribution of the Gini index for consecutive events of different intensities having light rain shows that as the WBGT intensity increases, the

distribution of the Gini index, although still peaking at 0.7–0.8, becomes narrower and more spiky. On both sides of the peak, the distribution shifts rightward. Different from the intensity, the entire distribution, including the peaks, of the Gini index distribution for different durations, shifts rightward as the number of days increases (Fig. 3b). Regardless, more intense and longer consecutive events occur more frequently under the more uneven distribution of light rain, and less frequently under less uneven light rain distribution. As the distribution of rain becomes more uneven, the composited consecutive events, on average, exhibit higher intensity and longer duration (Fig. 3c–f). Despite that, the WBGT on non-rainy days (black asterisk in Fig. 3c, d) is comparable to or slightly smaller than that in the largest Gini index cases, light rain days generally have larger WBGT than non-rainy days in most of the world (Supplementary Fig. 9a). Compared to too often light rainfall (a situation where every day has light rainfall with a Gini index approaching 0), occasional light rain

during consecutive humid heat waves (typically one to three days of light rain, Supplementary Fig. 10c) helps make the heat waves more intense and last longer. This can be explained by the correlation between surface latent heat flux and downward solar radiation, and the rate of latent heat flux change, respectively. The former indicates the extent of the control of evaporation by radiation (i.e., energy constraints[35,36]). When the Gini index is small, light rain occurs too frequently. There is a lack of intermittent sunshine to effectively evaporate surface water (i.e., high correlation coefficients). As a result, it weakens evaporation, leading to lower humid heat intensity. As the distribution of rain becomes more uneven, the correlation between latent heat flux and downward solar radiation decreases. In this case, water rather than energy is a limiting factor for surface evaporation[35,36]. Timely light rain enhances evaporation through increased soil moisture, resulting in higher humid heat intensity than the cases with more frequent light rain and the non-rainy cases both over the tropics and the mid-latitude regions. The rate of latent heat flux change over time, defined as the absolute value of the slope of linear regression of latent heat flux with time, reflects the stability of the change of latent heat flux during an event. The larger the change rate, the faster and more unstable the latent heat flux changes over time, and vice versa. Note that the change rate does not represent whether the latent heat flux becomes larger or smaller. This metric characterizes the ability of light rain to maintain near-surface water vapor through evaporation, which is of importance for the duration. As the distribution of rain becomes more uneven, the rate of latent heat flux change slows down (Fig. 3e, f), consistent with the changes in duration. This means that in a consecutive event, evaporation providing near-surface water vapor in intermittent non-rainy days following light rain days can be sustainable at a stable level for a longer duration.

To gain a more concrete understanding, we take some specific cases in different parts of the world as examples. During a humid heat wave in the grid cell where Shanghai is located (Supplementary Fig. 11a), evaporation on the first day with light rain is maintained at a consistently high level for the subsequent five days. After the humid heat wave lasts for six days, water vapor near the surface begins to diminish, while the second light rain on the seventh day makes evaporation continue to provide water vapor near the surface in the following two days. The uneven distribution of light rainfall occurrence during this consecutive event results in a smooth trend of latent heat flux changes, thus further prolonging the days of the humid heat wave. In other grid cells where densely populated cities like New Delhi, Kinshasa, and Sao Paulo are situated (Supplementary Fig. 11b–d), the situation is similar. Therefore, unevenly distributed light rain in time in consecutive events can sustain evaporation, which in turn maintains air humidity, further prolonging the duration of the humid heat waves and keeping their intensity at a high level. It is important to note that the harm of humid heat waves with higher intensity and longer duration is significant[37].

As extreme humid heat is primarily influenced by the amount of water vapor near the surface, which is largely generated through surface evaporation, two crucial factors come into play: downward solar radiation and wet surfaces. Interestingly, they often work in opposite directions under various weather conditions, and light rain proves to be the most effective in optimizing the interplay between downward solar radiation and wet surfaces for enhancing surface evaporation compared to both dry and heavily rainy cases. It achieves an ideal balance. However, it does not mean that the more frequent light rain, the longer the duration and the higher the intensity of consecutive humid heat waves. Figure 3 and Supplementary Fig. 11 manifest that the exacerbation of consecutive humid heat waves by light rain does not totally follow the condition of a single day. The key lies in several rain-free days during sporadic drizzle days. This allows the moisture accumulated on the surfaces following a day of light rain to have ample time to evaporate due to sustained radiation during the successive non-rainy days.

## Implications for simulated humid heat waves in GCMs

The findings in reanalysis/observations have important implications for simulated humid heat waves in Coupled Model Intercomparison Project Phase 5 and 6 (CMIP5&6) GCMs because they commonly suffer from "too much light rain and too little heavy rain"[27]. Since GCMs are used to project changes in humid heat waves, it is imperative to know the impact of excessive light rain on its simulation in the current climate before one can trust the GCM projections on future humid heat waves. Here, we take the National Center for Atmospheric Research Community Atmosphere Model version 5.3 (NCAR CAM5.3) as an example (see "Methods"). To reduce the overestimated frequency of light rain in this model, we employ a stochastic convective parameterization in CAM5 and conduct the simulations[25,32,38] (see "Methods"). As shown in Supplementary Fig. 12a, the simulation with the stochastic deep convection scheme (STOC) almost produces an identical occurrence frequency of different precipitation rates to that in observations compared to the default model simulation (CAM5)[26,28,32,38].

As light rain frequency is significantly reduced over most regions (Supplementary Fig. 12b), the Gini index of light rain mainly increases in low latitudes and decreases in mid-latitudes (Fig. 4a), corresponding to stronger WBGT in low latitudes and the opposite changes in mid-latitudes, respectively (Fig. 4b). The decrease of WBGT intensity in mid-latitudes is caused by the decrease in natural wet bulb temperature ($T_w$) due to the resulting decrease in evaporative water vapor (Supplementary Figs. 13 and 14). In low latitudes, the increase of WBGT intensity results from enhanced surface radiation, which leads to a significant increase in $T_w$ as well as increases in black globe temperature ($T_g$) and dry bulb temperature ($T_a$). The distinct changes in $T_w$ across latitudes result from a combination of the diverse background of light rain frequency and the different constraints on evaporation. In low latitudes where rainfall is extremely frequent (up to 75%, Fig. 1a) and is further overestimated by up to 20% in CAM5 (Supplementary Fig. 12b), consecutive days of light rain are common. As indicated by Seneviratne et al.[36], evaporation is mainly constrained by radiation characterized by the positive correlation coefficients between evaporation and radiation (Fig. 4c)[35,36]. After reducing excessive light rain in low latitudes, more consecutive days without rain[39] separate consecutive days of light rain, thus with more sustained radiation effectively increasing $T_w$ and mitigating its negative biases (Supplementary Fig. 13). In contrast, in mid-latitudes where rainfall is relatively infrequent (less than 45%, Fig. 1a), consecutive days without rain are common. Accordingly, evaporation there is mainly constrained by water, which is characterized by the positive correlation coefficients between evaporation and precipitation (Fig. 4c). Overestimated light rain in CAM5 separates consecutive non-rainy days, leading to positive biases of $T_w$ there. With the removal of the separated light rain days in mid-latitudes, surface water is not replenished timely following consecutive non-rainy days. As a result, the overestimated $T_w$ is alleviated. Note that all these changes, including surface latent heat flux, water vapor, and downward shortwave radiation, make the model simulation agree better with reanalysis and observations (Supplementary Figs. 13, 14 and Supplementary Table 1).

The explanation of the distinct changes in WBGT intensity across evaporation regimes after suppressing too frequent light rain globally is validated by the changes in the duration of consecutive events (Supplementary Fig. 15). The duration of consecutive events is longer in low latitudes where the intensities of WBGT and $T_w$ are stronger but shorter in middle latitudes where the intensities of WBGT and $T_w$ are weaker. The spatial distribution of duration changes resembles that of the changes of light rain frequency in consecutive events (i.e., the Gini index) (Fig. 4a), confirming their relationship as revealed in reanalysis and observations. Also, both of their simulations are improved as the bias of light rain is reduced (Supplementary Table 1). In CAM5, too frequent light rain leads to a small Gini index, resulting in more grid

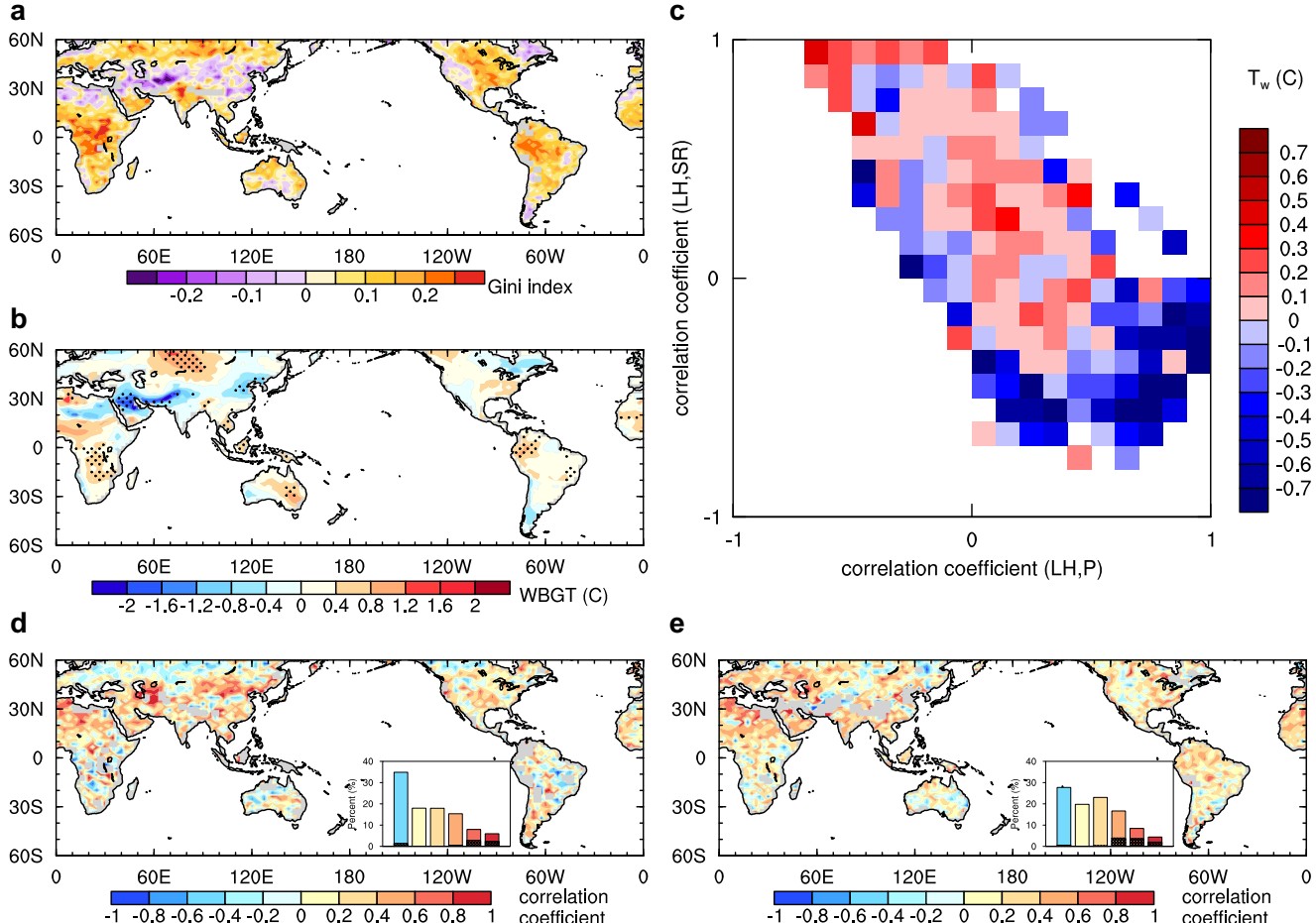

**Fig. 4 | Model simulation results. a, b** Differences between STOC and CAM5 (STOC minus CAM5) in simulating the Gini index of light rain distribution during and the day before consecutive events (**a**) and wet bulb globe temperature (WBGT) intensity (**b**) in the hottest four months. **c** Difference between STOC and CAM5 (STOC minus CAM5) in natural wet bulb temperature ($T_w$) in the hottest four months grouped by the correlation coefficients between surface latent heat flux (LH) and precipitation (P) and those between surface LH and surface downwelling shortwave radiation (SR). **d, e** Correlation coefficients between the average number of days of consecutive events and the corresponding Gini index in CAM5 (**d**) and STOC (**e**), respectively, the inset bar chart shows the percentage of the coefficients in the intervals of ≤ 0, 01–0.2, 0.2–0.4, 0.4–0.6, 0.6–0.8 and 0.8–1.0, exceeding the 95% confidence level of the *t* test are shaded. Differences in (**b**) statistically significant at the 95% confidence level are stippled.

cells featuring a negative correlation between the Gini index and the duration of events (Fig. 4d). In STOC, the correlation between the two is closer to the observations/reanalysis with the coefficient above 0.6 passing the significant test, showing a significant improvement in the simulation of the correlation between the Gini index and duration (Figs. 2c and 4e).

## Discussion

During humid heat waves, radiation-driven evaporation is usually difficult to maintain a stable level of near-surface water vapor as available surface water becomes depleted[30]. Timely light rain can supply water to the surface to allow the following sunshine (sometimes accompanied by light rain) to significantly increase surface latent heat flux despite the decrease of sensible heat flux. This results in an increase of the total enthalpy flux, especially over the arid and semi-arid regions (Supplementary Fig. 16). The spatial pattern of the total enthalpy flux changes resembles that of surface net radiation changes in which, as expected, the net longwave component increases and the net short-wave counterpart decreases. Despite the decreases of the total enthalpy fluxes in most humid regions, the decreases of planetary boundary layer height (PBLH) worldwide (Supplementary Fig. 16g) in humid heat days with light rain compared to those without rain can enhance WBGT as well. Although moderate to heavy rainfall can offer

more surface water than light rain and the associated PBLH is further decreased, the substantial decrease of downward radiation during rainfall effectively suppresses evaporation and thus the total enthalpy flux worldwide, terminating humid heat waves (Supplementary Fig. 17). Therefore, the amplification of humid heat waves by light rain is a tradeoff between water and energy constraints on evaporation.

This is crucial for consecutive humid heat waves where the common case is incorporating sporadic drizzle days amidst several rain-free days. This allows the moisture accumulated on the surfaces following a day of light rain, to have ample time to evaporate due to sustained radiation during the successive non-rainy days. This optimum strategy is confirmed by the model simulations. Over energy-limited evaporation regions with extremely frequent light rain, overestimated light rain largely makes successive non-rainy days following light rain absent, leading to underestimated natural wet bulb temperature and WBGT. In contrast, over water-limited evaporation regions with infrequent light rain, overestimated light rain can have more light rain days separate consecutive rain-free days, leading to overestimated natural wet bulb temperature and WBGT. Therefore, the overestimated frequency of light rain days in climate models may have different biases in future projections of humid heat waves in different regions[40], resulting in underestimated humid heat in energy-limited regions and over-estimated humid heat in water-limited regions.

As climate warms, heavy precipitation is projected to increase while light precipitation is expected to decrease[27]. Reduced light rain frequency during humid heat waves in the tropics may make humid heat waves more intense through increased radiation; The decrease of light rain during humid heat waves in mid-latitudes may reduce surface evaporation, which may not be sufficient to provide the moisture needed to sustain humid heat waves to some extent. The duration of humid heat waves in mid-latitudes may be shortened in the future. It suggests that the reduction in the frequency of light rain due to global warming in the future may greatly aggravate the severity of humid heat waves in the tropics where there are most developing economies more susceptible to humid heat waves. Therefore, the findings provide a crucial theoretical basis for understanding the impact of reduced light rain on humid heat waves under future climate warming.

## Methods

### Calculation of WBGT

The Wet Bulb Globe Temperature (WBGT) is a widely used international heat stress index (ISO 7243)[14,41]. This index is reliable and practical and is widely applied in evaluating outdoor heat stress to guide workers' activities, athletes and military training, etc.[42]. WBGT is a weighted average of the natural wet bulb temperature ($T_w$), black globe temperature ($T_g$), and dry bulb temperature ($T_a$):

$$WBGT = 0.7*T_w + 0.2*T_g + 0.1*T_a \qquad (1)$$

which incorporates the effects of temperature, humidity, radiation, and wind. Due to the complexity of measurements, WBGT is not a commonly used meteorological variable output by weather stations and models. Therefore, many studies have developed simplified forms of WBGT, such as[43–45]:

$$sWBGT = 0.567*T + 0.393*VP + 3.94 \qquad (2)$$

which only requires temperature and humidity and does not take into account variations in solar radiation and wind speed. It assumes moderate levels of radiation under light wind conditions, so sWBGT may have biases outside of this assumption.

In studying the impact of light rain on WBGT, we cannot ignore the effect of radiation. Therefore, we adopt the explicitly calculated WBGT in this study[16]. The Liljegren model[46] is used to calculate $T_w$ and $T_g$, which are obtained through iterative solutions using temperature, relative humidity, radiation, and wind speed (Eqs. 6 and 9 in ref. [46]). The natural wet bulb thermometer is fully exposed to the environment and can represent the effect of solar radiation and wind speed on the human body's ability to cool down through perspiration. The black globe thermometer reflects the influence of surface radiation. Kong[16] implemented the Liljegren model in Python for the calculation of $T_w$ and $T_g$.

### The non-uniformity of precipitation distribution

The Gini index is commonly used in economics to measure income inequality in a country or group[47,48]. It is also widely applied in various other fields such as education, health, ecology, and agriculture[49–51]. In meteorology, the Gini index can be used to evaluate the temporal uniformity of daily precipitation distribution[34]. To investigate the uneven distribution of precipitation during consecutive humid heat waves, we calculate the Gini index of precipitation to characterize its unevenness[34] within a consecutive heat event:

$$Gini\ index = \frac{1}{n}\left(n+1-2\left(\frac{\sum_{i=1}^{n}(n+1-i)y_i}{\sum_{i=1}^{n}y_i}\right)\right) \qquad (3)$$

where $n$ is the number of days in the consecutive event, and $y_i$ is the daily precipitation. The Gini index ranges from 0 to 1, with the smaller value indicating a more even temporal distribution of precipitation, i.e., 0 represents evenly distributed rain throughout the consecutive event, and the larger value indicating a more discrete distribution, i.e., 1 represents an event with precipitation on only one day. Therefore, the Gini index is sensitive to the number of rainy days during a consecutive heat event.

### Reanalysis and observations

We use the fifth-generation global atmospheric reanalysis data of the European Center for Medium-Range Weather Forecasts (ERA5)[52] to calculate WBGT. Hourly WBGT is computed using hourly 2 m temperature, 2 m dew point temperature, 10 m wind, surface air pressure, and radiations with a spatial resolution of $0.25_{\circ}$ (January 1, 2001-December 31, 2020). Using the Global Precipitation Measurement mission (GPM) daily precipitation data[53] for the same period (January 1, 2001-December 31, 2020), the light rain frequency is calculated with a spatial resolution of $0.25_{\circ}$ Surface latent heat flux is estimated using 20-year daily evaporation data from the Global Land Evaporation Amsterdam Model (GLEAM) version 3.7[54,55], which is based on satellite and reanalysis data from 1980 to 2020, with a spatial resolution of $0.25_{\circ}$ We use the GLEAM products during 2001–2020. The Clouds and the Earth's Radiant Energy System (CERES) project provides satellite-based earth radiation data, providing monthly mean radiation data with a spatial resolution of $1_{\circ}$ We use downwelling shortwave radiation flux data from CERES Energy Balanced and Filled (EBAF) Edition 4.1 (Ed4.1) product[56] from January 2001 to December 2020.

### Global climate model with light rain suppressed

The GCM used in this study is the National Center for Atmospheric Research (NCAR) Community Atmosphere Model version 5.3 (CAM5.3)[57]. The horizontal resolution is $0.9_{\circ}\times 2.5_{\circ}$ and the vertical resolution is 30 levels from the surface to 3.6 hPa. The convective parameterization scheme is the Zhang-McFarlane scheme[58] with dilute convective available potential energy modification[59]. The Plant and Craig stochastic deep convection parameterization scheme[60] is adjusted and incorporated into the Zhang-McFarlane scheme[25,32,38], and the frequency of light rain at the daily scale can be greatly reduced[28,32,61]. Two Atmospheric Model Intercomparison Project experiments are conducted using the standard CAM5 and introducing the stochastic convective scheme (STOC), respectively, with prescribed, seasonally varying climatological present-day (1982–2001 mean) sea surface temperatures and sea ice extent, recycled yearly as the lower boundary conditions. The experiments are simulated for 6 years, and the last 5 years are used for analysis.

## Data availability

ERA5 reanalysis data sets are available from https://cds.climate.copernicus.eu/cdsapp#!/dataset/reanalysis-era5-pressure-levels?tab=form. GPM data can be obtained at https://pmm.nasa.gov/data-access/downloads/gpm. GLEAM data can be downloaded from https://www.gleam.eu. CERES EBAF Edition 4.1 data are available at https://ceres.larc.nasa.gov/data/#energy-balanced-and-filled-ebaf. The model simulation data in this study have been deposited in Zenodo (https://doi.org/10.5281/zenodo.13189627).

## Code availability

The CESM1.2.1-CAM5.3 source code can be downloaded from the CESM official website: http://www2.cesm.ucar.edu. The stochastic deep-convection code is available from https://doi.org/10.5281/zenodo.4543261. The relevant code for each figure has been deposited in Zenodo (https://doi.org/10.5281/zenodo.13189627).

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

## Acknowledgements
Y.W. is supported by the National Key Research and Development Program of China Grant 2022YFF0802002, the National Natural Science Foundation of China Grant 41975126, and the Tsinghua University Initiative Scientific Research Program Grant 20223080041. G.J.Z. is supported by the US National Science Foundation grant AGS-2054697 and the US Department of Energy (DOE), Office of Science, Biological and Environmental Research Program (BER), under Award Number DE-SC0022064. We thank the European Center for Medium-Range Weather Forecasts (ECMWF) and the National Aeronautics and Space Administration (NASA) for making the reanalysis and observational products available.

## Author contributions
Y.W. and G.Z. conceived and designed the research. Z.Z. and Y.W. performed the analysis. Y.W. conducted the model simulation. Z.Z., Y.W., and G.Z. interpreted the results. Z.Z. and Y.W. and G.Z. wrote the paper. Z.Z., Y.W., G.Z., C.X., W.X., and M.Y. discussed the results and edited the manuscript.

## Competing interests
The authors declare no competing interests.
