## [Peer Review File · Nature Communications]

Light Rain Exacerbates Extreme Humid HeatEditorial Note: Parts of this Peer Review File have been redacted as indicated to remove third-party material where no permission to publish could be obtained.

Reviewers' comments:

Reviewer #1 (Remarks to the Author):

Review of "Light Rain Exacerbates Extreme Humid Heat" by Zhang et al.

This paper investigates the role of rainfall in controlling the intensity and duration of humid heat extremes across the global tropics and mid-latitudes. A combination of reanalysis, observations and climate model simulations are used to show that rainy days, interspersed with dry days can increase the intensity and duration of humid heat events. There are some regional differences because humid heat at lower latitudes are more radiation limited but mid-latitudes are more moisture limited. The paper goes on to consider the representation of rainfall in global climate models and demonstrates that a model with less light (erroneous) rainfall represents humid heat extremes differently.

Whilst there is existing literature that links humid heat extremes with rainy days, no paper has done this comprehensively across the globe, so for this reason the findings are novel. Indeed, there is a distinct lack of literature on the drivers of humid heat extremes, so this paper provides an excellent addition to understanding. The paper is generally well written, with good structure and clear figures. For all these reasons I recommend to publish once my major and minor revisions below have been addressed.

Major comments

My most major comment on this paper is the classification of 'light rain' days and how these are considered alongside heavier rain days. Firstly, the paper considers 'light rain', which is defined as rainfall 0.1-20 mm/day. These events are also described as "drizzle days." (e.g. lines 24-25 and 204). I would not consider a day with 20mm of rainfall to be light rainfall nor drizzle, especially at mid-latitudes. I think you should consider changing this terminology, including in the title of the paper. You should also justify why an upper threshold of 20mm/day was chosen (regardless of how it is described) and consider how sensitive your results are to the chosen threshold. Secondly, I was unsure how you are dealing with the days that have more than 20mm/day, compared with days with no rainfall. Are these two categories considered separately or together? How often is the rainfall higher than 20mm/day in each region and what happens to humid heat on these days? Line 132-134 may go some way to address this point but if it does, it is not explicit enough regarding heavy events. Another example is Lines 282-284: "Although moderate to heavy rainfall can offer more surface water than light rain, the substantial decrease of downward radiation during rainfall effectively suppresses evaporation and significantly cools the air, terminating extreme humid heat waves." I don't see direct evidence for this in the paper (or any other paper referenced).

My second major comment is about how consistent ERA5 T, Td, wind etc is with the GLEAM latent heat flux and GPM rainfall observations. ERA5 is itself produced using a model with parameterised convection. Does ERA5 also have too much light rainfall? I suspect it does. How might this impact the results? Are the rainy days in ERA5 consistent with the observed rainy days?

My third major comment is about how much the STOC experiment has improved the representation of heavy/light rainfall days. In the section " Implications for simulated humid heat

waves in GCMs”, the STOC experiment clearly changes the amount of light rainfall but is this actually closer to reality, and if so how much of an improvement is it? I would like to see some evidence of this, for example histograms/pdfs of rainfall in the two experiments vs GPM.

Minor comments

Line 26-27 “By **reducing too much light rain** in a global climate model..” this phrasing is awkward and I didn’t understand it until I re-read the opening paragraph after I’d read the whole paper. Suggest to rephrase.

Fig 1b (and elsewhere) – it is not clear what is plotted here. You say ‘WBGT intensity’ but do you just mean average WBGT over the 4 hottest months? I don’t see a specific calculation of intensity in your methods section.

Fig 1 caption “Probability of extreme **humid heat days or the day before** with light rain.” And also line 99-100 “The ratios of the standard deviation of the **extreme humid heat days or the day before with light rain to** that of extreme humid heat waves exceed” And probably elsewhere, this wording is confusing and I don’t understand what you have done exactly. How many days prior to the heatwave are you looking for a rainy day? Is it same day and previous day only? Please reword and/or make clearer in the methods.

Lines 164-166 “It suggests that **not too often light rain** during consecutive humid heat waves (typically one to three days of light rain, Supplementary Figure S6) helps make the heat waves more intense and last longer.” What does ‘not too often’ mean specifically? It is poor wording anyway but I am unsure how often is ‘not too often’.

Fig 3c – the black asterisk is not commented on in the text. If I have understood correctly, it shows that on no rain days the WBGT is actually on average the highest compared to all other days with some rainfall. Does this fit with the rest of your results? Is this due to all the very arid, hot areas such as the Sahara? as the black asterisks are not so high in Fig S8.

Fig S9 – it would be better to plot the rainfall accumulations as blue bars behind the red and black lines, so the reader knows how much rainfall there was (rather than just label with arrows)

Line 196-209: This paragraph is worded as though you have shown this statistically for the whole globe, but I think these conclusions are based only on Fig S9, which is just a few case studies. It is ok to do this but I would tone down the language to be less sure in your statements. E.g. “Following our analysis of 4 events it appears that....”

Fig 4 shows the between STOC and CAM5. Be clear about whether this is STOC minus CAM5 or CAM5 minus STOC (I assume latter).

Fig 4 caption “d, e, Same as Fig. 2c except for CAM5 (d) and STOC (e).” it would be easier for the reader if you just said what these panels showed in the Fig 4 caption.

Line 214-242 “Evaporation **there** [low latitudes] is mainly constrained by radiation characterized by the positive correlation coefficients between evaporation and radiation (Fig. 4c)” But this panel does not separate the data by latitude as far as I can see – what are you using to come to this conclusion?

Birch et al. (2022) Future changes in African heatwaves and their drivers at the convective scale, <https://doi.org/10.1175/JCLI-D-21-0790.1> is one of the few papers that considers the different drivers (including rainfall) of humid heat equatorial, tropical and subtropical areas (esp figs 13+14) and the effect of climate models with different representations of convection. Your results are consistent with that paper, so I would recommend referencing it in your intro and/or conclusions.

Discussion – it is probably worth re-emphasising the fact that CMIP5/6 type models use parameterised convection, which has too much light rainfall, so will bias future projections of humid heat and the biases vary between overestimations and underestimations in different regions.

Reviewer #2 (Remarks to the Author):

Review for: "Light Rain Exacerbates Extreme Humid Heat".

This manuscript argues that light rain makes humid heat stronger and humid heatwaves longer. The physical pathway for this to happen is described as: light rain prior to humid heat provides surface water availability which enhances surface evaporation when the sun comes out in the following days; sporadic light rain events within a multi-day humid heatwave tend to prolong the events by replenishing soil moisture. The physical process outlined above is interesting and the investigation into it is helpful for illuminating the physical drivers of humid heat. However, I do not think the aforementioned hypotheses are well supported by the results. The authors frequently make causality statements like "light rain exacerbates humid heat", but what they show is only correlations at summer average scale. To me, this correlation is more likely due to the control of external factors like monsoon dynamics rather than the influence of light rain on humid heat. I believe more in-depth diagnoses are needed to convince readers the physical connection between light rain and humid heat. Therefore, I suggest rejecting this manuscript to leave the authors more time for improvement, but I do encourage them to resubmit to Nature Communication.

Major concern:

As outlined above, my major concern is that the results shown in this manuscript cannot support a causality relation between light rain and humid heat. Below are some examples:

The fact that the equator and Southeast Asia have both high WBGT and frequent light rain (Fig. 1 a and b) simply reflect they are climatologically hot-humid rather than telling anything about the effects of light rain on WBGT values.

The positive correlation between "the occurrence frequency of light rain and WBGT intensity" (presumably at summer average scale) in Fig. 1c much more likely reflects the inter-annual variation of monsoon dynamics or other large-scale processes (the red areas in Fig. 1c are consistently monsoon regions). When the monsoon is strong, both WBGT and precipitation are likely to increase as a result of stronger moisture transport. In fact, if the author re-produce Fig. 1b-c using total precipitation, I strongly suspect they will get the same pattern. Similar argument applies to Fig. 1f.

Similarly, the fact that "more than 50% of humid heat waves are accompanied by light rain" (Fig. 1e) could be simply a result of the hot-humid climate or the influence of moisture transport.

The authors found a positive correlation between precipitation Gini index and WBGT values, and suggest that the uneven distribution of light rain within humid heatwaves will make the events stronger and longer. However, I don't think this is helpful for supporting the hypothesis that light rain prolongs heatwave by replenishing soil moisture during the events. It may simply reflect that people rarely see a heatwave that rains everyday. Also, from a purely statistical point of view, we should expect a higher Gini index for a long heatwave, since there will be only one or two rainy days within a

long period, say 10 days. Imagine a two-day heatwave, the most uneven distribution you can get is simply one day raining and one day rain-free. But more importantly, I don't think the even or uneven distribution of precipitation should be a focus here given the ultimate goal is to understand whether light rain within heatwaves will make the events stronger and/or longer.

Finally, comparing Fig. 1c, Fig. 2c (or Fig. S4c), and Fig. 4b, the spatial distribution is largely inconsistent among the regions showing a positive light rain-WBGT correlation (Fig. 1c), regions where large precipitation Gini index associates with stronger and longer heatwave (Fig. 2c or Fig. S4c), and regions showing reduced WBGT after correcting the "too much light rain" bias (Fig. 4b). To me, it really suggests that these analyses are not capturing the physical connection between light rain and humid heat (if this connection does exist).

In general, the authors should be very cautious about making causality arguments since they are dealing with correlations in reanalysis. More in-depth diagnosis is needed to convince people of the potential influence of light rain on humid heat. It may be beneficial to select representative regions and look into the progression of actual heatwave events. The authors only select a single heatwave over 4 grid points. It may be better to do a composite analysis in order to be more representative. Meanwhile, I suggest looking at multiple variables during the progression of heatwaves including temperature, humidity, soil moisture, surface fluxes, radiations, boundary layer height, etc. Please see my comments below for why it's useful to look at these variables.

Meanwhile, it might be helpful to look at moderate or heavy rain as well which can be compared against light rain.

Other concerns:

Please distinguish between natural wet-bulb temperature (T_{nw}) and wet-bulb temperature (T_w).

The authors argue that light rain increases humid heat by increasing evaporation and humidity. However, temperature will decrease if more energy goes to evaporation. Given the opposite response of temperature and humidity, the authors do not explain why we should expect a net increase in heat stress. If one is using wet-bulb temperature (T_w) which is essentially moist enthalpy, a simple surface energy repartition between latent and sensible heat will not change T_w as long as their sum remains constant. In fact, since WBGT places more weights on temperature compared with T_w , WBGT should decrease if more energy is used for evaporation. This means that other processes in addition to surface energy partition are needed to explain the potential positive correlation between light rain (soil moisture) and T_w /WBGT. One possible mechanism is the response of boundary layer growth and dry air entrainment. With more energy used for evaporation, the boundary layer becomes shallower which traps both sensible and latent heat flux into a smaller volume and also reduces dry air entrainment from free-troposphere. These boundary layer responses will enhance heat stress. Please check the recent published paper (<https://journals.ametsoc.org/view/journals/clim/aop/JCLI-D-23-0132.1/JCLI-D-23-0132.1.xml>) on

the coupling between soil moisture and T_w .

It's useful to discuss the implication of "light rain increasing humid heat" on model simulations given that models tend to produce "too much light rain". However, I'm not sure to what extent the difference between CAM5 and STOC simulations can be attributed to light rain frequency changes. Switching the convection scheme may induce lots of changes including cloud, radiation, the overall precipitation, and even circulations.

The writing needs to be improved. Right now, there are multiple places that are quite confusing.

Reply to the comments by Reviewer #1

We thank the reviewer for the comments and suggestions to improve our manuscript. Below are our point-by-point responses to these comments. The reviewer's comments are in italics, our responses are in normal font, and manuscript revisions are in blue.

This paper investigates the role of rainfall in controlling the intensity and duration of humid heat extremes across the global tropics and mid-latitudes. A combination of reanalysis, observations and climate model simulations are used to show that rainy days, interspersed with dry days can increase the intensity and duration of humid heat events. There are some regional differences because humid heat at lower latitudes are more radiation limited but mid-latitudes are more moisture limited. The paper goes on to consider the representation of rainfall in global climate models and demonstrates that a model with less light (erroneous) rainfall represents humid heat extremes differently.

Whilst there is existing literature that links humid heat extremes with rainy days, no paper has done this comprehensively across the globe, so for this reason the findings are novel. Indeed, there is a distinct lack of literature on the drivers of humid heat extremes, so this paper provides an excellent addition to understanding. The paper is generally well written, with good structure and clear figures. For all these reasons I recommend to publish once my major and minor revisions below have been addressed.

Reply: We thank the reviewer for the positive remarks.

Major comments

1. *My most major comment on this paper is the classification of 'light rain' days and how these are considered alongside heavier rain days. Firstly, the paper considers 'light rain', which is defined as rainfall 0.1-20 mm/day. These events are also described as "drizzle days." (e.g. lines 24-25 and 204). I would not consider a day with 20mm of rainfall to be light rainfall nor drizzle, especially at mid-latitudes. I think you should consider changing this terminology, including in the title of the paper. You should also justify why an upper threshold of 20mm/day was chosen (regardless of how it is described) and consider how sensitive your results are to the chosen threshold. Secondly, I was unsure how you are dealing with the days that have more than 20mm/day, compared with days with no rainfall. Are these two categories considered separately or together? How often is the rainfall higher than 20mm/day in each region and what happens to humid heat on these days? Line 132-134 may go some way to address this point but if it does, it is not explicit enough regarding heavy events. Another example is Lines 282-284: "Although moderate to heavy rainfall can offer more surface water than light rain, the substantial decrease of downward radiation during rainfall effectively suppresses evaporation and significantly cools the air, terminating extreme humid heat waves." I don't see direct evidence for this in the paper (or any other paper referenced).*

Reply: We thank the reviewer for these insightful comments. Light rain defined as daily rainfall ranging from 0.1 to 20 mm d⁻¹ follows the criterion provided by previous studies (e.g., Na et al., 2020). Considering the reviewer’s concerns of 20 mm daily rainfall in mid-latitudes being too high for light rain and the sensitivity of the conclusions to the definition of light rain, we reduced the upper limit of light rain from 20 mm d⁻¹ to 10 mm d⁻¹ following another study (Qian et al., 2007). We are pleased to report that the results of using the two criteria for defining light rain are very similar (Fig. R1), reaffirming our conclusions. Here we take the temporal correlation coefficients between light rain probability and WBGT during the hottest four months for example.

Fig. R1. Temporal correlation coefficients between light rain probability and WBGT during the hottest four months. Light rain is defined as daily rainfall ranging from 0.1 to 10 mm d⁻¹ (a) and from 0.1 to 20 mm d⁻¹ (b), respectively. Areas exceeding the 95% confidence level of the t-test are stippled.

In the revision, the definition of light rain has been changed to daily rainfall ranging from 0.1 to 10 mm d⁻¹ in Lines 69-70 and all the figures have been updated:

We analyze the distribution of the occurrence frequency of light rain (0.1 - 10 mm d⁻¹) (Qian et al., 2007) at the daily timescale and its link with heat stress index WBGT (see Methods).

Days with more than 20 mm d⁻¹ of rainfall (Na et al., 2020; Cui et al., 2022) and days without rainfall are considered separately. The occurrence frequency of daily rainfall intensity larger than 20 mm d⁻¹ is much smaller (maxima < 20%, Fig. R2a) (Wang et al., 2016). The correlation between the daily rainfall intensity larger than 20 mm d⁻¹ and WBGT is not significant (Fig. R2b), indicating that WBGT is not regulated by the frequency of moderate to heavy rain. On extreme humid heat days and the day before,

the occurrence with daily rainfall intensity larger than 20 mm d^{-1} is relatively rare, with maxima $< 10\%$ (Fig. R3). This is because downward solar radiation and resulting air temperature in moderate to heavy rain days are lower than those in light rain days (Fig. R4).

In the revision, Fig. R2 is included as Fig. S1a and b. The related discussion is added in Lines 78-83:

Compared with the light rain frequency, the frequency of moderate to heavy rain with daily rainfall intensity greater than 20 mm d^{-1} (Na et al., 2020; Cui et al., 2022) is much smaller showing maxima smaller than 20% (Supplementary Figure S1a) (Wang et al., 2016), and its correlation with WBGT is not significant (Supplementary Figure S1b), indicating that WBGT is not regulated by the frequency of moderate to heavy rainfall. In this regard, the following analyses focus on the comparison between light-rain cases and no-rain cases.

Fig. R2. (a) Global distributions of occurrence frequency of daily rainfall larger than 20 mm d^{-1} and (b) temporal correlation coefficients between daily rainfall larger than 20 mm d^{-1} and WBGT during the hottest four months. Areas exceeding the 95% confidence level of the t-test in b are stippled.

Fig. R3. Probability of extreme humid heat days and the day before with daily rainfall larger than 20 mm d^{-1} .

Fig. R4. Differences in downward solar radiation (a) and 2-m air temperature (b) between moderate-to-heavy rain days and light rain days in humid heat waves.

References:

- Na, Y., Fu, Q., & Kodama, C. (2020). Precipitation Probability and Its Future Changes From a Global Cloud-Resolving Model and CMIP6 Simulations. *Journal of Geophysical Research: Atmospheres*, *125*(5), e2019JD031926. <https://doi.org/10.1029/2019JD031926>
- Qian, W., Fu, J., & Yan, Z. (2007). Decrease of light rain events in summer associated with a warming environment in China during 1961–2005. *Geophysical Research Letters*, *34*(11). <https://doi.org/10.1029/2007GL029631>
- Wang, Y., Zhang, G. J., & Craig, G. C. (2016). Stochastic convective parameterization improving the simulation of tropical precipitation variability in the NCAR CAM5. *Geophysical Research Letters*, *43*(12), 6612–6619. <https://doi.org/10.1002/2016GL069818>

Cui, Z., Wang, Y., Zhang, G. J., Yang, M., Liu, J., & Wei, L. (2022). Effects of Improved Simulation of Precipitation on Evapotranspiration and Its Partitioning Over Land. *Geophysical Research Letters*, 49(5). <https://doi.org/10.1029/2021GL097353>

2. *My second major comment is about how consistent ERA5 T, Td, wind etc is with the GLEA M latent heat flux and GPM rainfall observations. ERA5 is itself produced using a model with parameterised convection. Does ERA5 also have too much light rainfall? I suspect it does. How might this impact the results? Are the rainy days in ERA5 consistent with the observed rainy days?*

Reply: ERA5 hourly 2-m air temperature, 2-m dew point temperature, 10-m winds, surface air pressure, and radiation are used to calculate WBGT. These variables are produced by assimilating their direct observations. Thus, their fidelity should not be degraded by the biases of ERA5 precipitation resulting from cloud parameterization schemes (e.g., convection and cloud microphysics schemes). The reviewer is right that the ERA5 precipitation, although constrained by these assimilated variables, is largely a model parameterization product. We do not use ERA5 precipitation in the analysis but employ GPM precipitation instead. Therefore, the light rain bias in ERA5 (Lavers et al., 2022) does not affect the results of this study.

Reference:

Lavers, D. A., Simmons, A., Vamborg, F., & Rodwell, M. J. (2022). An evaluation of ERA5 precipitation for climate monitoring. *Quarterly Journal of the Royal Meteorological Society*, 148(748), 3152–3165. <https://doi.org/10.1002/qj.4351>

3. *My third major comment is about how much the STOC experiment has improved the representation of heavy/light rainfall days. In the section “Implications for simulated humid heat waves in GCMs”, the STOC experiment clearly changes the amount of light rainfall but is this actually closer to reality, and if so how much of an improvement is it? I would like to see some evidence of this, for example histograms/pdfs of rainfall in the two experiments vs GPM.*

Reply: Actually, the STOC simulation drastically improved the precipitation pdf, including the light rain occurrence frequency (Fig. R5). Since we have documented the precipitation improvement and its associated effects on aerosol, radiative forcing and land-atmosphere interaction in several previous publications (Wang et al., 2016, 2017, 2021; Cui et al., 2022), we just added the brief discussion regarding the improvement of the simulation of rainfall PDF (Fig. R5, current Supplementary Fig. S16) in Lines 259-262 in the revised manuscript:

As shown in Supplementary Figure S16, the simulation with the stochastic deep convection scheme (STOC) almost produces an identical occurrence frequencies of different precipitation rates as in observations compared to the default model simulation (CAM5) (Wang et al., 2016, 2017, 2021; Cui et al., 2022).

Fig. R5. The PDFs of rainfall intensity over land in the hottest 4 months over (60°S–60°N).

References:

- Cui, Z., Wang, Y., Zhang, G. J., Yang, M., Liu, J., & Wei, L. (2022). Effects of Improved Simulation of Precipitation on Evapotranspiration and Its Partitioning Over Land. *Geophysical Research Letters*, 49(5). <https://doi.org/10.1029/2021GL097353>
- Wang, Y., Zhang, G. J., & Craig, G. C. (2016). Stochastic convective parameterization improving the simulation of tropical precipitation variability in the NCAR CAM5. *Geophysical Research Letters*, 43(12), 6612–6619. <https://doi.org/10.1002/2016GL069818>
- Wang, Y., Zhang, G. J., & He, Y.-J. (2017). Simulation of Precipitation Extremes Using a Stochastic Convective Parameterization in the NCAR CAM5 Under Different Resolutions. *Journal of Geophysical Research: Atmospheres*, 122(23), 12,875–12,891. <https://doi.org/10.1002/2017JD026901>
- Wang, Y., Xia, W., Liu, X., Xie, S., Lin, W., Tang, Q., et al. (2021). Disproportionate control on aerosol burden by light rain. *Nature Geoscience*, 14(2), 72–76. <https://doi.org/10.1038/s41561-020-00675-z>

Minor comments

1. Line 26-27 “By reducing too much light rain in a global climate model..” this phrasing is awkward and I didn’t understand it until I re-read the opening paragraph after I’d read the whole paper. Suggest to rephrase.

Reply: In Lines 28-29 in the revised manuscript, we have rephrased it:

Current global climate models (GCMs) commonly simulate excessive light rain. After reducing this bias in a GCM, ...

2. Fig 1b (and elsewhere) – it is not clear what is plotted here. You say ‘WBGT intensity’ but do you just mean average WBGT over the 4 hottest months? I don’t see a specific calculation of intensity in your methods section.

Reply: Yes, it is the average of WBGT over the 4 hottest months. In Lines 73-74 in the revision, we defined the WBGT intensity:

where WBGT intensities (defined as the average of WBGT over the 4 hottest months) are high as well (Fig. 1b).

3. *Fig 1 caption “Probability of extreme humid heat days or the day before with light rain.” And also line 99-100 “The ratios of the standard deviation of the extreme humid heat days or the day before with light rain to that of extreme humid heat waves exceed” And probably elsewhere, this wording is confusing and I don’t understand what you have done exactly. How many days prior to the heatwave are you looking for a rainy day? Is it same day and previous day only? Please reword and/or make clearer in the methods.*

Reply: We analyze light rain occurrence during periods of extreme humidity and the day prior to such events. For example, if there is a 3-day spell of intense humid heat, we will analyze the likelihood of light rain over 4 consecutive days, encompassing the 3-day event and the day preceding it. To avoid confusion, we reword it in Lines 101-102 in the revision:

The probability of light rain occurring on any given day during the extreme humid heat events and the day before such events ...

4. *Lines 164-166 “It suggests that not too often light rain during consecutive humid heat waves (typically one to three days of light rain, Supplementary Figure S6) helps make the heat waves more intense and last longer.” What does ‘not too often’ mean specifically? It is poor wording anyway but I am unsure how often is ‘not too often’.*

Reply: We thank the reviewer for the comment. When we mention “too often light rainfall”, what we mean is a situation when there is light rain every day (i.e., with a Gini index approaching 0) during consecutive humid heat waves. Otherwise, it is not too often. In Lines 194-197 in the revision, we clarified it:

Compared to too often light rainfall (a situation when there is light rain every day, with a Gini index approaching 0), occasional light rain during consecutive humid heat waves (typically one to three days of light rain, Supplementary Figure S12) helps make the heat waves more intense and last longer.

5. *Fig 3c – the black asterisk is not commented on in the text. If I have understood correctly, it shows that on no rain days the WBGT is actually on average the highest compared to all other days with some rainfall. Does this fit with the rest of your results? Is this due to all the very arid, hot areas such as the Sahara? as the black asterisks are not so high in Fig S8.*

Reply: We thank the reviewer for pointing out this. It can be seen from Fig. S10a in the revision that in consecutive events, WBGT on light rain days is generally higher than that on non-rainy days in most of the world, consistently supporting that light rain

aggravates consecutive humid heat waves compared to non-rainy events. When separating into (20°S-20°N) and (20°N-60°N) latitudes (Fig. S14 in the revision), the comparable WBGT on non-rainy days with that in the largest Gini index cases is seen in the synthesis of arid/semi-arid and humid regions over (20°N-60°N). In the revised manuscript, we commented on this in Lines 191-194:

Despite the comparable WBGT on non-rainy days (black asterisk in Fig. 3c) with that in the largest Gini index cases, light rain days generally have larger WBGT than non-rainy days in most of the world (Supplementary Figure S10a).

6. Fig S9 – it would be better to plot the rainfall accumulations as blue bars behind the red and black lines, so the reader knows how much rainfall there was (rather than just label with arrows)

Reply: Following the suggestion, we have updated Fig. S15 in the revised manuscript (here Fig. R6).

Fig. R6. a-d, The changes in latent heat flux and WBGT over time in a consecutive humid heat wave in the grid cells where Shanghai (a), New Delhi (b), Kinshasa (c), and Sao Paulo (d) are located, respectively. The bar denotes the daily amount of rainfall.

7. Line 196-209: This paragraph is worded as though you have shown this statistically for the whole globe, but I think these conclusions are based only on Fig S9, which is just a few case studies. It is ok to do this but I would tone down the language to be less sure in your statements. E.g. “Following our analysis of 4 events it appears that....”

Reply: We make these conclusions based on Fig. 3 while Fig. S15 in the revision just shows some specific cases for a more concrete understanding. Following the suggestion, we explicitly point out that the discussion is based on Figs. 3 and S15 in Lines 233-237 in the revision.

However, it does not mean that the more light rain, the longer the duration and the higher the intensity of consecutive humid heat waves. Figs. 3, S14 and S15 figure out

that the exacerbation of consecutive humid heat waves by light rain does not totally follow the condition of a single day. The key lies in several rain-free days during sporadic drizzle days.

8. *Fig 4 shows the between STOC and CAM5. Be clear about whether this is STOC minus CAM5 or CAM5 minus STOC (I assume latter).*

Reply: The difference is STOC minus CAM5 because STOC has less light rain. We clarified this in Lines 302-307 in the revision.

a, b, Differences between STOC and CAM5 (STOC minus CAM5) in simulating the Gini index of light rain distribution during and the day before consecutive events (**a**) and WBGT intensity (**b**) in the hottest four months. **c,** Difference between STOC and CAM5 (STOC minus CAM5) in natural wet bulb temperature (T_w) in the hottest four months grouped by the correlation coefficients between surface latent heat flux (LH) and precipitation (P) and those between surface LH and surface downwelling shortwave radiation (SR).

9. *Fig 4 caption “d, e, Same as Fig. 2c except for CAM5 (d) and STOC (e).” it would be easier for the reader if you just said what these panels showed in the Fig 4 caption.*

Reply: Done (Lines 307-312 in the revision).

d, e, Correlation coefficients between the average number of days of consecutive events and the corresponding Gini index in CAM5 (**d**) and STOC (**e**), respectively, the inset bar chart shows the percentage of the coefficients in the intervals of ≤ 0 , 0-0.2, 0.2-0.4, 0.4-0.6, 0.6-0.8 and 0.8-1.0, exceeding the 95% confidence level of the t-test are shaded. Differences in **b** statistically significant at the 95% confidence level are stippled.

10. *Line 214-242 “Evaporation there [low latitudes] is mainly constrained by radiation characterized by the positive correlation coefficients between evaporation and radiation (Fig. 4c)” But this panel does not separate the data by latitude as far as I can see – what are you using to come to this conclusion?*

Reply: This has been informed by Fig.8a (here Fig. R7) in the cited reference 36 (Seneviratne et al., 2010). It can be seen that in low latitudes, especially over tropical rainforests there are positive correlation coefficients between evaporation and radiation. To avoid confusion, the sentence in Line 274 in the revision has been reworded as follows:

As indicated by Seneviratne et al. (2010), evaporation there is ...

[REDACTED]

Fig. R7. Estimation of the drivers of evapotranspiration (moisture and radiation) based on simulations from the GSWP project (from Fig. 8a in Seneviratne et al., 2010).

Reference:

Seneviratne, S. I., Corti, T., Davin, E. L., Hirschi, M., Jaeger, E. B., Lehner, I., et al. (2010). Investigating soil moisture–climate interactions in a changing climate: A review. *Earth-Science Reviews*, 99(3), 125–161. <https://doi.org/10.1016/j.earscirev.2010.02.004>

11. Birch et al. (2022) Future changes in African heatwaves and their drivers at the convective scale, <https://doi.org/10.1175/JCLI-D-21-0790.1> is one of the few papers that considers the different drivers (including rainfall) of humid heat equatorial, tropical and subtropical areas (esp figs 13+14) and the effect of climate models with different representations of convection. Your results are consistent with that paper, so I would recommend referencing it in your intro and/or conclusions.

Reply: We thank the reviewer for bringing this paper to our attention. It has been cited in Lines 333-335 in the revision:

the overestimated frequency of light rain days in the climate models may have different biases on future projections of humid heat waves in different regions (Birch et al., 2022).

12. Discussion – it is probably worth re-emphasising the fact that CMIP5/6 type models use parameterised convection, which has too much light rainfall, so will bias future projections of humid heat and the biases vary between overestimations and underestimations in different regions.

Reply: We thank the reviewer for the helpful suggestion. It has been recalled in Lines 333-336 in the revision:

Therefore, the overestimated frequency of light rain days in climate models may have different biases on future projections of humid heat waves in different regions (Birch et al., 2022), resulting in underestimated humid heat in energy-limited regions and overestimated humid heat in water-limited regions.

Reply to the comments by Reviewer #2

We thank the reviewer for the comments and suggestions to improve our manuscript. Below are our point-by-point responses to these comments. The reviewer's comments are in italics, our responses are in normal font, and manuscript revisions are in blue.

This manuscript argues that light rain makes humid heat stronger and humid heatwaves longer. The physical pathway for this to happen is described as: light rain prior to humid heat provides surface water availability which enhances surface evaporation when the sun comes out in the following days; sporadic light rain events within a multi-day humid heatwave tend to prolong the events by replenishing soil moisture. The physical process outlined above is interesting and the investigation into it is helpful for illuminating the physical drivers of humid heat. However, I do not think the aforementioned hypotheses are well supported by the results. The authors frequently make causality statements like "light rain exacerbates humid heat", but what they show is only correlations at summer average scale. To me, this correlation is more likely due to the control of external factors like monsoon dynamics rather than the influence of light rain on humid heat. I believe more in-depth diagnoses are needed to convince readers the physical connection between light rain and humid heat. Therefore, I suggest rejecting this manuscript to leave the authors more time for improvement, but I do encourage them to resubmit to Nature Communication.

Reply: We thank the reviewer for the positive remarks, and appreciate the reviewer's concerns regarding the correlation of light rain occurrence and humid heat extremes being due to other common controlling factors. Besides performing correlation analysis, we also carry out composite analysis and provide modeling evidence to demonstrate their physical relationship. In addition, we further provide evidence showing that monsoon dynamics does not simultaneously regulate the variabilities of precipitation and humid heat. Please see our detailed responses to the comments below. Furthermore, we should point out that this study intends to show that light rain exacerbates extreme humid heat, and we do not argue that light rain causes humid heat waves.

Major concern

As outlined above, my major concern is that the results shown in this manuscript cannot support a causality relation between light rain and humid heat. Below are some examples:

- 1. The fact that the equator and Southeast Asia have both high WBGT and frequent light rain (Fig. 1 a and b) simply reflect they are climatologically hot-humid rather than telling anything about the effects of light rain on WBGT values.*

The positive correlation between "the occurrence frequency of light rain and WBGT intensity" (presumably at summer average scale) in Fig. 1c much more likely reflects the inter-annual variation of monsoon dynamics or other large-scale processes (the red

areas in Fig. 1c are consistently monsoon regions). When the monsoon is strong, both WBGT and precipitation are likely to increase as a result of stronger moisture transport. In fact, if the author re-produce Fig. 1b-c using total precipitation, I strongly suspect they will get the same pattern. Similar argument applies to Fig. 1f.

Similarly, the fact that “more than 50% of humid heat waves are accompanied by light rain” (Fig. 1e) could be simply a result of the hot-humid climate or the influence of moisture transport.

Reply: We understand the reviewer’s concerns about the correlation between light rain and humid heat (i.e., Fig. 1c and f in the revision). The reviewer suggested that maybe they were both governed by interannual variations of monsoon dynamics or other large-scale processes. Following the reviewer’s suggestion, we performed an analysis of the temporal correlation between total precipitation and WBGT (Fig. R1). In all monsoonal regions, the correlation coefficients are much smaller than those between light rain frequency and WBGT (Fig. 1c in the revision), even having negative values in many regions, such as southeast Asia, part of India and part of Amazonia. This is likely because although more precipitation can provide more water for evapotranspiration to increase near-surface moisture, the reduced radiation suppresses evapotranspiration and reduces sensible heat flux to heat the near-surface air (Fig. R2).

Fig. R1. Temporal correlation coefficients between total precipitation and WBGT in the hottest four months, and areas exceeding the 95% confidence level of the t-test are stippled.

Fig. R2. Differences in downward solar radiation (a) and 2-m air temperature (b) between moderate-to-heavy rain days and light rain days in humid heat waves.

The physical processes of the incidence of precipitation and humid heat are different. Monsoonal precipitation (especially in terms of precipitation amount) relies on moisture transport in the atmospheric column rather than the near-surface moisture changes while the latter determines humid heat via the land-atmosphere interactions in the planetary boundary layer (PBL) (e.g., evapotranspiration) (Brouillet and Jousaume, 2019). As a result, monsoon dynamics does not simultaneously regulate the variabilities of precipitation and humid heat.

In the revision, Fig. R1 is included in the supplementary as Fig. S1c and more discussion is added in Lines 84-87:

Note that the temporal correlation between total precipitation and WBGT is insignificant or even negative, especially over monsoon regions (Supplementary Figure S1c). This indicates that the correlation between light rain and humid heat is not a result of temporal variation of monsoon dynamics or other large-scale processes.

References:

Brouillet, A., & Jousaume, S. (2019). Investigating the Role of the Relative Humidity in the Co-Occurrence of Temperature and Heat Stress Extremes in CMIP5 Projections. *Geophysical Research Letters*, 46(20), 11435–11443. <https://doi.org/10.1029/2019GL084156>

2. The authors found a positive correlation between precipitation Gini index and WBGT values, and suggest that the uneven distribution of light rain within humid heatwaves will make the events stronger and longer. However, I don't think this is

helpful for supporting the hypothesis that light rain prolongs heatwave by replenishing soil moisture during the events. It may simply reflect that people rarely see a heatwave that rains everyday. Also, from a purely statistical point of view, we should expect a higher Gini index for a long heatwave, since there will be only one or two rainy days within a long period, say 10 days. Imagine a two-day heatwave, the most uneven distribution you can get is simply one day raining and one day rain-free. But more importantly, I don't think the even or uneven distribution of precipitation should be a focus here given the ultimate goal is to understand whether light rain within heatwaves will make the events stronger and/or longer.

Reply: In this study, the Gini index of precipitation is only applied to consecutive humid heat waves that last three or more consecutive days. Along with the day before the event, there are at least four days to temporally distribute light rain. As mentioned in the original manuscript, given that more than 50% of humid heat waves are accompanied by light rain (Fig. 1e in the revision), this may be incorrectly interpreted as consecutive events having more frequent (i.e. nearly continuous) light rain. However, by employing the Gini index of precipitation (a combination of the number of rainy days and their uneven distribution), it is found that a more uneven distribution of light rain within consecutive humid heatwaves makes the events stronger and last longer. Furthermore, the uneven temporal distribution of light rain in consecutive humid heatwaves further supports that light rain and humid heat are not regulated by the same common factor such as large-scale monsoon systems. In this regard, we feel it is important to discuss this in a separate section. In the revision, more discussion and clarifications are added in Lines 171-177:

Given that more than 50% of humid heat waves are accompanied by light rain (Fig. 1e), this may be incorrectly interpreted as consecutive events having more frequent or nearly continuous light rain. However, by employing the Gini index of precipitation (a combination of the number of rainy days and their distribution), it is found that a more uneven distribution of light rain within consecutive humid heatwaves will actually make the events stronger and last longer. The uneven temporal distribution of light rain in consecutive humid heatwaves further confirms that light rain and humid heat are not regulated by the same common factor such as large-scale monsoon systems.

3. *Finally, comparing Fig .1c, Fig. 2c (or Fig. S4c), and Fig. 4b, the spatial distribution is largely inconsistent among the regions showing a positive light rain-WBGT correlation (Fig .1c), regions where large precipitation Gini index associates with stronger and longer heatwave (Fig. 2c or Fig. S4c), and regions showing reduced WBGT after correcting the “too much light rain” bias (Fig. 4b). To me, it really suggests that these analyses are not capturing the physical connection between light rain and humid heat (if this connection does exist).*

Reply: For consecutive humid heatwaves, the modeling result is consistent with that in observations/reanalysis. As shown in Fig. S12 in the original manuscript (Fig. S20 in the revised manuscript), regions with increases in the Gini index of light rain (Fig. 4a

in the revision) all experience longer consecutive events. This consistency is also reflected in Figs. 4d and 4e in the revision, showing an improvement in the simulated correlation between the Gini index of light rain distribution and the average number of days of consecutive events after reducing excessive light rain.

As for the seemingly inconsistent distributions between Fig. 1c and Fig. 4b in the revision, the reasons have been discussed in Lines 231-251 in the original manuscript. Here we recall it again. It should be mentioned first that evaporation is constrained by both available downward radiation and water at the surface. Among different regions, the dominant drivers of evaporation differ (Fig.8a in the cited reference 36, Seneviratne et al., 2010) (here Fig. R3). In Fig. 1c, the strong positive correlation coefficients between evaporation and precipitation are mainly located in water-limited regions of evaporation while weak and negative correlation coefficients between evaporation and radiation are located in radiation-limited regions of evaporation. Keeping this feature in mind, the physical consistency in Fig. 4b can be established. The distribution in WBGT changes resembles that of the duration changes of consecutive humid heatwaves (Fig. S20 in the revision). As light rain frequency is significantly reduced (Fig. S17 in the revision), the intensity of warm season WBGT (mainly natural wet bulb temperature, T_w) is reduced in water-limited regions and increased in radiation-limited regions (Fig. 4b). The distinct changes in WBGT across different evaporation regimes result from a combination of the diverse background of light rain frequency and the different constraints. In radiation-limited regions where rainfall is extremely frequent (up to 75%, Fig. 1a in the revision) and is further overestimated by up to 20% in CAM5 (Fig. S17 in the revision), consecutive days of light rain are common cases. After reducing excessive light rain there, more consecutive days without rain separate consecutive days of light rain, thus with more sustained radiation effectively increasing T_w and mitigating its negative biases (Figs. S18 and S19 in the revision). In contrast, in water-limited regions where rainfall is relatively infrequent (less than 45%, Fig. 1a), consecutive days without rain are common cases. Overestimated light rain in CAM5 separates consecutive non-rainy days, leading to positive biases of T_w there. With the removal of the separated light rain days in water-limited regions, surface water is not replenished timely following consecutive non-rainy days. As a result, the overestimated T_w is alleviated.

In the revision, to avoid confusion, the original Fig. 4a is moved to the supplementary as Fig. S17 and is replaced by the original Fig. S12b and more clarifications are made in Lines 263-266.

As light rain frequency is significantly reduced over most regions (Supplementary Figure S17), the Gini index of light rain mainly increases in low latitudes and decreases in mid-latitudes (Fig. 4a), corresponding to stronger WBGT in low latitudes and the opposite changes in mid-latitudes, respectively (Fig. 4b).

[REDACTED]

Fig. R3. Estimation of the drivers of evapotranspiration (moisture and radiation) based on simulations from the GSWP project (from Fig. 8a in Seneviratne et al., 2010).

Reference:

Seneviratne, S. I., Corti, T., Davin, E. L., Hirschi, M., Jaeger, E. B., Lehner, I., et al. (2010). Investigating soil moisture–climate interactions in a changing climate: A review. *Earth-Science Reviews*, 99(3), 125–161. <https://doi.org/10.1016/j.earscirev.2010.02.004>

4. *In general, the authors should be very cautious about making causality arguments since they are dealing with correlations in reanalysis. More in-depth diagnosis is needed to convince people of the potential influence of light rain on humid heat. It may be beneficial to select representative regions and look into the progression of actual heatwave events. The authors only select a single heatwave over 4 grid points. It may be better to do a composite analysis in order to be more representative. Meanwhile, I suggest looking at multiple variables during the progression of heatwaves including temperature, humidity, soil moisture, surface fluxes, radiations, boundary layer height, etc. Please see my comments below for why it's useful to look at these variables.*

Reply: Following the reviewer's suggestions, we conducted a composite analysis on humid heatwaves with light rain and without rain to demonstrate whether their background large-scale meteorological fields are significantly different. Taking India, eastern China, the eastern United States and Europe as examples (Fig. R4), the background large-scale meteorological fields (i.e., surface pressure, 500 hPa geopotential height and 10-m wind speed) are divided into two groups according to the criterion of whether the selected grid cell (asterisks in Fig. R4) experiencing humid heat waves have light rain or without rain. It is shown that over all the regions, there are no significant differences between the two groups. Therefore, the very similar large-scale meteorological conditions in the two groups support the conclusion that light rain exacerbates humid heat.

Figs. 1c, 1f, 2c, and S5, etc. in the revision are analyzed at the interannual time scale. Below, we further select one year (e.g., 2001, note that the results for other years are similar) to do the composite analysis to screen out the possible influence of interannual

variabilities of large-scale dynamical processes on both light rain and humid heat. Fig. R5 shows the PDF of averaged WBGT and duration for both light rain and non-rainy cases. The higher percentage of stronger WBGT and longer duration when there is light rain confirm that the occurrence of light rain exacerbates humid heatwaves.

Fig. R4. The composite fields of surface pressure (shadings), 500hpa geopotential height (contours) and 10m wind speed (vectors) during humid heatwaves with light rain (left) and without rain (right) occurring in the selected grid cells (asterisks) in India (a, b), eastern China (c, d), eastern United States (e, f) and Europe (g, h).

Fig. R5. The occurrence frequency of WBGT intensity (a) and duration (b) of humid heat events for light-rain and non-rainy cases over (60°S –60°N) in year 2001.

In the revision, Figs. R4 and R5 are included as Figs. S7-S9. The related discussion is added in Lines 117-128 and 140-142:

To screen out possible influences of interannual variabilities of large-scale dynamical processes on both light rain and humid heat, we perform a composite analysis of humid heat events within a year. Taking 2001 as an example (the results for other years are similar, figure not shown), Supplementary Figure S7 shows the probability distribution function of averaged WBGT for both light rain and non-rainy cases. There is a higher percentage of stronger WBGT when there is light rain. The large-scale dynamical background during humid heatwaves with light rain and without rain are almost identical, characterized by the similar large-scale meteorological fields (i.e., surface pressure, 500hpa geopotential height and 10m wind speed) between the two groups (Supplementary Figure S8), which is divided by whether the selected grid cell on humid heat waves has light rain or without rain. These two figures along with Supplementary Figure S1c support the conclusion that light rain exacerbates humid heat rather than that they co-vary with the variation of large-scale dynamical processes.

Given that more light rain days in the warm season are associated with more humid heat waves, some of these humid heat waves are a result of consecutive humid heat waves prolonged by the incidence of light rain (Supplementary Figure S9).

5. *Meanwhile, it might be helpful to look at moderate or heavy rain as well which can be compared against light rain.*

Reply: We thank the reviewer for the insightful comment. Days with more than 20 mm d^{-1} of rainfall (Na et al., 2020; Cui et al., 2022) and days without rainfall are considered separately. The occurrence frequency of daily rainfall intensity larger than 20 mm d^{-1} is much smaller (maxima < 20%, Fig. R6a) (Wang et al., 2016). The correlation between the daily rainfall intensity larger than 20 mm d^{-1} and WBGT is not significant (Fig. R6b), indicating that WBGT is not regulated by the frequency of moderate to heavy rain. On extreme humid heat days and the day before, the occurrence of daily rainfall intensity larger than 20 mm d^{-1} is relatively rare (maxima < 10%) (Fig. R7). This is because downward solar radiation and resulting air temperature in moderate to heavy rain days are lower than those in light rain days (Fig. R2).

In the revision, Fig. R6 is included as Fig. S1a and b. The related discussion is added in Lines 78-83:

Compared with the light rain frequency, the frequency of moderate to heavy rain with daily rainfall intensity greater than 20 mm d^{-1} (Na et al., 2020; Cui et al., 2022) is much smaller showing maxima smaller than 20% (Supplementary Figure S1a) (Wang et al., 2016), and its correlation with WBGT is not significant (Supplementary Figure S1b), indicating that WBGT is not regulated by the frequency of moderate to heavy rainfall. In this regard, the following analyses focus on the comparison between light-rain cases and no-rain cases.

Fig. R6. (a) Global distributions of occurrence frequency of daily rainfall larger than 20 mm d^{-1} and (b) temporal correlation coefficients between daily rainfall larger than 20 mm d^{-1} and WBGT during the hottest four months. Areas exceeding the 95% confidence level of the t-test in b are stippled.

Fig. R7. Probability of extreme humid heat days and the day before with daily rainfall larger than 20 mm d^{-1} .

References:

- Na, Y., Fu, Q., & Kodama, C. (2020). Precipitation Probability and Its Future Changes From a Global Cloud-Resolving Model and CMIP6 Simulations. *Journal of Geophysical Research: Atmospheres*, *125*(5), e2019JD031926. <https://doi.org/10.1029/2019JD031926>
- Wang, Y., Zhang, G. J., & Craig, G. C. (2016). Stochastic convective parameterization improving the simulation of tropical precipitation variability in the NCAR CAM5. *Geophysical Research Letters*, *43*(12), 6612–6619. <https://doi.org/10.1002/2016GL069818>
- Cui, Z., Wang, Y., Zhang, G. J., Yang, M., Liu, J., & Wei, L. (2022). Effects of Improved Simulation of Precipitation on Evapotranspiration and Its Partitioning Over Land. *Geophysical Research Letters*, *49*(5). <https://doi.org/10.1029/2021GL097353>

Other concerns

1. Please distinguish between natural wet-bulb temperature (T_{nw}) and wet-bulb temperature (T_w).

Reply: We thank the reviewer for pointing this out. In the manuscript, T_w is used to represent the natural wet bulb temperature required for calculating WBGT (Liljegren et al., 2008; Kong and Huber, 2022). T_w is measured by a sensor embedded in a wet wick exposed to the environment, taking into account the effects of solar radiation and wind. The measured wet bulb temperature, often referred to as "natural wet bulb temperature," represents the cooling process of the human body through perspiration, and it can be considered a specific case of wet bulb temperature. In the revision, we corrected "wet-bulb temperature" to "natural wet-bulb temperature".

References:

- Liljegren, J. C., Carhart, R. A., Lawday, P., Tschopp, S. & Sharp, R. Modeling the Wet Bulb Globe Temperature Using Standard Meteorological Measurements. *Journal of Occupational and Environmental Hygiene* **5**, 645–655 (2008).

Kong, Q. & Huber, M. Explicit Calculations of Wet-Bulb Globe Temperature Compared With Approximations and Why It Matters for Labor Productivity. *Earth's Future* **10**, (2022).

2. *The authors argue that light rain increases humid heat by increasing evaporation and humidity. However, temperature will decrease if more energy goes to evaporation. Given the opposite response of temperature and humidity, the authors do not explain why we should expect a net increase in heat stress. If one is using wet-bulb temperature (T_w) which is essentially moist enthalpy, a simple surface energy repartition between latent and sensible heat will not change T_w as long as their sum remains constant. In fact, since WBGT places more weights on temperature compared with T_w , WBGT should decrease if more energy is used for evaporation. This means that other processes in addition to surface energy partition are needed to explain the potential positive correlation between light rain (soil moisture) and T_w /WBGT. One possible mechanism is the response of boundary layer growth and dry air entrainment. With more energy used for evaporation, the boundary layer becomes shallower which traps both sensible and latent heat flux into a smaller volume and also reduces dry air entrainment from free-troposphere. These boundary layer responses will enhance heat stress. Please check the recent published paper (<https://journals.ametsoc.org/view/journals/clim/aop/JCLI-D-23-0132.1/JCLI-D-23-0132.1.xml>) on the coupling between soil moisture and T_w .*

Reply: We thank the reviewer for the valuable comment. If we neglect the associated adjustments of net shortwave and longwave radiation at the surface when evaporation changes in response to altered soil moisture by rainfall, as the reviewer stated, the surface enthalpy flux is conserved. However, the feedback of net surface radiation fluxes will break this rule. Below is our reasoning. The surface energy budget is approximated as:

$$R_s - R_l - LE - SH = 0 \quad (R1)$$

or

$$R_s - R_l - c_p * c_h * U * (T_s - T_a) - L * c_e * U * (q_s - q_a) = 0 \quad (R2)$$

where R_s , R_l , LE and SH are net shortwave radiation, net longwave radiation, latent heat flux and sensible heat flux, respectively, T_s and T_a are the temperatures of the surface and near-surface atmosphere respectively, q_s and q_a are the specific humidity of the surface and near-surface atmosphere respectively, L is the latent heat of evaporation, c_p is the specific heat capacity of air at constant pressure, U is the wind speed, and c_h and c_e are the turbulent exchange coefficients for sensible and latent heat fluxes at the surface respectively, with their values approximately equal. Increasing surface moisture will increase evaporation, but evaporative cooling will decrease T_s , resulting in a decrease in sensible heat flux. If the net change in enthalpy flux is zero, then the air above will have no energy gain.

On the other hand, neglecting changes in albedo, we assume that R_s remains constant when q_s changes, according to the Stefan-Boltzmann law, the net longwave radiation can be approximated as:

$$R_l = a * \sigma * T_s^4 \quad (R3)$$

where a simultaneously accounts for the effects of non-blackbody radiation and downward longwave radiation from the atmosphere, and σ is the Stefan-Boltzmann constant. Therefore, the change of surface energy budget Eq. R2 when q_s changes can be expressed as:

$$-a * \sigma * 4 * T_s^3 \frac{dT_s}{dq_s} - c_p * c_h * U * \frac{dT_s}{dq_s} - L * c_e * U = 0 \quad (R4)$$

If changes in R_l are neglected, that is, setting the first term on the lhs of Eq. R4 to zero, and $c_h \approx c_e$, it can be deduced that:

$$\frac{dT_s}{dq_s} = -\frac{L}{c_p} \quad (R5)$$

That is, surface evaporative cooling balances the increase in soil moisture. However, after considering the changes in R_l , we obtain:

$$\frac{dT_s}{dq_s} > -\frac{L}{c_p} \quad (R6)$$

This implies that evaporative cooling is less than the increase in soil moisture due to rainfall. Physically, when soil moisture increases, evaporation will increase, leading to an increase in air humidity. The evaporative cooling at the surface lowers the surface temperature T_s , reducing the sensible heat flux, and the near-surface air temperature T_a decreases. If there are no other changes in the atmosphere, the two are balanced (Eq. R5). However, due to the cooler surface, longwave cooling will be reduced. As a result, the actual decrease in T_s will be less than that without changes in R_l . This will lead to a positive net enthalpy flux into the air (Eq. R6), increasing the natural wet bulb temperature. The shortwave albedo feedback can make an additional positive contribution if more soil moisture makes the surface appear darker, as it will absorb more sunlight, increasing T_s and thus the natural wet bulb temperature. Also, we thank the reviewer for bringing our attention to the paper by Kong and Huber (2023) on changes in the planetary boundary layer thickness contributing to the increased wet bulb temperature. In the revised manuscript, Kong and Huber (2023) is cited in Lines 52-53, and the above discussion is added in Lines 319-320:

When humid heat waves occur, the soil is wetter than normal because the planetary boundary layer is shallower (Kong and Huber 2023) and the accumulation of water vapor near the surface increases humidity.

This mechanism can be verified by a mathematical description (see Methods).

3. *It's useful to discuss the implication of "light rain increasing humid heat" on model simulations given that models tend to produce "too much light rain". However, I'm not sure to what extent the difference between CAM5 and STOC simulations can*

be attributed to light rain frequency changes. Switching the convection scheme may induce lots of changes including cloud, radiation, the overall precipitation, and even circulations.

Reply: Yes, as the reviewer stated, switching the convection scheme indeed alters clouds, radiation, precipitation and circulations. The impacts of the stochastic deep convection scheme on these aspects have been evaluated in our previous papers (e.g., Wang et al., 2016, 2018; Wang and Zhang, 2016). However, since “excessive light rain” is a well-known issue among the CMIP5&6 models and reduced light rain must be accompanied by less cloud and more shortwave radiation reaching the surface, we highlight the role of light rain. In fact, we are concerned with the changes in humid heat resulting from changes in light rain days on which light rain and the associated surface shortwave radiation both regulate surface evapotranspiration.

In the revision (Lines 259-262), we rephrase the sentences (Lines 228-230 in the original manuscript):

As shown in Supplementary Figure S16, the simulation with the stochastic deep convection scheme (STOC) almost produces an identical occurrence frequency of different precipitation rates as in observations compared to the default model simulation (CAM5) (Wang et al., 2016, 2017, 2021; Cui et al., 2022).

and make revisions in Lines 333-336 in the revised manuscript:

Therefore, the overestimated frequency of light rain days in the climate models may have different biases on future projections of humid heat waves in different regions (Birch et al., 2022), resulting in underestimated humid heat in energy-limited regions and overestimated humid heat in water-limited regions.

References:

- Wang, Y., Zhang, G. J., & Craig, G. C. (2016). Stochastic convective parameterization improving the simulation of tropical precipitation variability in the NCAR CAM5. *Geophysical Research Letters*, 43(12), 6612–6619. <https://doi.org/10.1002/2016GL069818>
- Wang, Y., Zhang, G. J., & Jiang, Y. (2018). Linking Stochasticity of Convection to Large-Scale Vertical Velocity to Improve Indian Summer Monsoon Simulation in the NCAR CAM5. *Journal of Climate*, 31(17), 6985–7002. <https://doi.org/10.1175/JCLI-D-17-0785.1>
- Wang, Y., & Zhang, G. J. (2016). Global climate impacts of stochastic deep convection parameterization in the NCAR CAM 5. *Journal of Advances in Modeling Earth Systems*, 8(4), 1641–1656. <https://doi.org/10.1002/2016MS000756>

4. The writing needs to be improved. Right now, there are multiple places that are quite confusing.

Reply: Thank the reviewer for the suggestion. We have further refined the writing in the revision.

REVIEWER COMMENTS

Reviewer #2 (Remarks to the Author):

Review for “Light Rain Exacerbates Extreme Humid Heat”

I appreciate that the authors properly addressed most of my comments, and the manuscript is clearly improved. My only major concern left is that more evidence needs to be provided to better establish the physical connection between light rain and humid heat. Please see my detailed suggestions below. I recommend publishing this work after the following major and minor concerns are addressed.

Major concern:

The authors argue that light rain exacerbates humid heat through replenishing soil moisture, sustaining evaporation while not reducing incoming solar radiation too much. The argument sounds reasonable. Some evidence is shown to support this argument mainly in Figure S3 where a positive (negative) correlation between light rain and latent heat flux (solar radiation) is demonstrated. However, this is not enough, and can be improved in the following two ways:

1) Instead of only looking at latent heat flux and incoming solar radiation, it's suggested to also examine sensible heat flux, downward and upwelling longwave radiation, surface reflected solar radiation and boundary layer height.

WBGT and T_w is affected by both latent and sensible heat flux. Although latent flux increases with light rain, sensible heat flux decreases. How about the total enthalpy flux? Also to explain the enthalpy flux change, one needs to look at all components of the surface radiative fluxes. As the authors also mentioned in the methods, although incoming solar radiation reduces, net longwave radiation may increase because of a more humid atmosphere and a cooler surface. Can the increase in net longwave radiation outweigh the decrease in solar radiation, leading to a net enhancement in enthalpy flux? Show the evidence. In addition, as mentioned in my first-round comments, boundary layer height can be an important variable to look at as it determines the volume that surface enthalpy flux is spreaded in. For example, even if net enthalpy flux remains the same between humid heat days with light rain and no rain, a shallower boundary layer under light rain days may still enhance T_w and WBGT.

2) It's useful to repeat the analyses above for humid heat without rain, with light rain, and with moderate or heavy rain. By analyzing the full surface turbulent and radiative fluxes, and boundary layer depth across three types of humid heat, the authors can more effectively establish that light rain is on a sweet spot for exacerbating humid heat. The authors argue that moderate or heavy rain reduces incoming solar radiation too much and thus diminishes humid heat. Although Figure S21 compares solar radiation between light rain and moderate-to-heavy rain days, how about longwave radiation and surface enthalpy flux? Is the solar radiation reduction in moderate-to-heavy rain days strong enough to cause significant reductions in enthalpy flux? Is it possible that the evapotranspiration becomes energy limited in moderate-to-heavy rain days, so that more water

replenishment won't further raise evaporation (or change evaporative fraction)?

Minor concerns:

(1) I appreciate the “mathematical description of the impact of light rain on T_w ” in the methods. However, the derivation is problematic. In deriving Eq. 6, the authors assume that both dT_a/dq_s and dq_a/dq_s are zero, which is wrong. We should expect $dT_a/dq_s < 0$ and $dq_a/dq_s > 0$.

When R_s , R_l and conductance parameters in Eq. (4) are held constant, the correct interpretation is that the sum of latent and sensible flux remain invariant to q_s changes. Namely, the moist enthalpy gradient between the surface and the air should be constant in order to maintain a constant enthalpy flux. But we cannot know whether moist enthalpy at the surface or the air will change or not. That is to say, surface energy balance alone does not offer enough constraint. Physically, moist enthalpy in the air could still be altered by other processes like advection, boundary layer depth response, mixed-layer top entrainment, etc. Land-atmosphere coupling makes sure surface and air moist enthalpy always change at the same rate.

If one assumes all these other processes are not affected by q_s , then it's reasonable to believe that moist enthalpy at both surface and the air won't change. But, this is not the case in the real world. For example, boundary layer growth will be suppressed under wet soil with less buoyancy flux. This will also change entrainment flux at mixed-layer top.

(2) The authors argue that light rain not only makes humid heat longer but also more intense. However, this seems not to be supported by the results. In figure S6, humid heat intensity is slightly weaker with light rain across the majority of the world. In Fig. 3c and Fig. S14a-b, WBGT intensity without rain is also higher than that with rain disregarding the Gini index. So, it seems that light rain will prolong humid heat but has little impact on or even tend to reduce humid heat intensity.

(3) By showing the Gini index, the authors demonstrate that when a humid heat wave is prolonged by light rain, it's sporadic rain among rain-free days rather than raining every day. This is helpful! But the authors should avoid making statements like “a higher Gini index will make a humid heat wave longer or stronger”. That's because the definition of Gini index inherently depends on the heatwave duration. Imagine a precipitation event that lasts two days, the Gini index will be lower if it happens within a 3-day heatwave and higher if it is embedded within a 10-day heatwave. The positive correlation between Gini index and duration is more likely to reflect such inherent dependence of Gini index calculation on duration, rather than suggesting a higher Gini index makes heat waves longer. My point is it's useful to describe the phenomena, i.e. the uneven distribution of rain, or make hypotheses based on that. But they should avoid going too far making causality arguments that cannot be supported by the results.

(4) In Line 191, the authors argue that “for given intensities and durations, the higher the Gini index, the higher the frequency of consecutive events” which is not the case in Fig. 3. The event frequency first increases with the Gini index and then decreases. It simply reflects the shape of the distribution of Gini index: lower in both tails and higher in-between. It's most obvious in terms of the group of duration. For longer duration, the Gini index frequency peaks at a higher index value, and

vice versa. Given this interpretation, I'm not sure how Fig. 3 a-b is useful.

(5) Why is it useful to show that the inter-annual variation of total humid heat days and the days with light rain match well in figure S5? It's not surprising that a major proportion of humid heat days vary consistently with the total humid heat days. How is this information useful?

(6) I suggest removing Eq. 1-2 and referring readers to Liljegren et al. (2008) for the details. Eq. 1-2 is rearranged from the energy balance of the WBGT sensor. But this rearrangement also makes it difficult to interpret intuitively (the authors didn't do it either). So, adding Eq. 1-2 won't help readers understand but introducing lots of unnecessary notations.

(7) Line 527: change implement to implemented.

(8) In Figure 3d, and Figure S14c-d, should the right y-axis be negative?

(9) In Figure S2 and S18, multiplying the 0.7, 0.2 and 0.1 weights on T_w , T_g and T_a makes it hard to distinguish the color scale. Better to show the original value.

(10) Line 51: move citation 20 to the end of the sentence.

(11) There are confusions on the methodological details at several places. Below are some examples:

In figure 1c, is the correlation calculated for summer average WBGT and light rain frequency across the 20-year study period? Similar confusion exists for other places when correlation is calculated.

Line 102: How is the 95th percentile calculated? Is it calculated across the whole summer or conditional on some intervals around the target calendar day?

In line 109, the authors state that the correlation is calculated between "the number of days with light rain" and "the number of days with extreme humid heat". However, the caption of figure 1f seems to state that the former quantity should be "the number of humid heat days or the day before" with light rain. Please clarify and check other places to avoid similar confusion.

In figure 1f, why calculate the standard deviations of humid heat days with or without light rain? How to interpret the ratio between these two standard deviations? What does it mean by the "ratio exceeds 0.5 in most regions and approaches 1 in the Amazon"? These seem confusing to me.

Reply to the comments by Reviewer #2

We thank the reviewer for the comments and suggestions to improve our manuscript. Below are our point-by-point responses to these comments. The reviewer's comments are in italics, our responses are in normal font, and manuscript revisions are in blue.

I appreciate that the authors properly addressed most of my comments, and the manuscript is clearly improved. My only major concern left is that more evidence needs to be provided to better establish the physical connection between light rain and humid heat. Please see my detailed suggestions below. I recommend publishing this work after the following major and minor concerns are addressed.

Reply: We thank the reviewer for the positive remarks.

Major concern

The authors argue that light rain exacerbates humid heat through replenishing soil moisture, sustaining evaporation while not reducing incoming solar radiation too much. The argument sounds reasonable. Some evidence is shown to support this argument mainly in Figure S3 where a positive (negative) correlation between light rain and latent heat flux (solar radiation) is demonstrated. However, this is not enough, and can be improved in the following two ways:

1) Instead of only looking at latent heat flux and incoming solar radiation, it's suggested to also examine sensible heat flux, downward and upwelling longwave radiation, surface reflected solar radiation and boundary layer height.

WBGT and T_w is affected by both latent and sensible heat flux. Although latent flux increases with light rain, sensible heat flux decreases. How about the total enthalpy flux? Also to explain the enthalpy flux change, one needs to look at all components of the surface radiative fluxes. As the authors also mentioned in the methods, although incoming solar radiation reduces, net longwave radiation may increase because of a more humid atmosphere and a cooler surface. Can the increase in net longwave radiation outweigh the decrease in solar radiation, leading to a net enhancement in enthalpy flux? Show the evidence. In addition, as mentioned in my first-round comments, boundary layer height can be an important variable to look at as it determines the volume that surface enthalpy flux is spreaded in. For example, even if net enthalpy flux remains the same between humid heat days with light rain and no rain, a shallower boundary layer under light rain days may still enhance T_w and WBGT.

Reply: We thank the reviewer for the constructive suggestions. This investigation into the land surface energy balance requires that all the variables be self-consistent. Unfortunately, no such data is available although some of the individual components from different sources are available. If collecting them from multiple observations, the surface energy balance will not be guaranteed, let alone the substantial discrepancies among different observations. Similarly, while the reanalysis data can provide all the

variables, they are not constrained by the surface energy balance either due to data assimilation. Therefore, here we use these variables from the STOC simulation. Following the suggestion, surface net radiation fluxes and the shortwave and longwave components, total enthalpy fluxes as well as sensible and latent components, and the planetary boundary layer height (PBLH) on humid heat days with light rain and without rain are shown (Fig. R1). As speculated, on light rain days, latent heat fluxes increase (Fig. R1d) and sensible heat fluxes decrease (Fig. R1e). The total enthalpy fluxes increase mainly over arid and semi-arid regions, where surface available water is the limiting factor for evapotranspiration (Fig. R1f), thus increasing natural wet-bulb temperature (Fig. S6a in the revision). The spatial pattern of the total enthalpy flux changes resembles that of surface net radiation changes (Fig. R1c) in which, as expected, the net longwave component increases (Fig. R1b) and the net shortwave counterpart decreases (Fig. R1a). Despite the decreases of the total enthalpy fluxes in most humid regions, the decreases of PBLH worldwide (Fig. R1g) in humid heat days with light rain compared to those without rain, as the reviewer previously suggested, can enhance WBGT as well. Therefore, in this revision, we have included some discussions on the changes of PBLH.

As noted by the reviewer's minor comment 2, light rain prolongs humid heat more efficiently compared to enhancing the intensity. In this situation, the intensity of the total enthalpy flux change cannot work for this exacerbation. Instead, the rate of change of latent heat flux is used in analysis (Fig. 3e and f in the revision).

In the revision, Fig. R1 is included as Fig. S16. The related discussion is added in Lines 332-341:

Timely light rain can supply water to the surface to allow the following sunshine (sometimes accompanied by light rain) to significantly increase surface latent heat flux despite the decrease of sensible heat flux. This results in an increase of the total enthalpy flux, especially over the arid and semi-arid regions (Supplementary Figure S16). The spatial pattern of the total enthalpy flux changes resembles that of surface net radiation changes in which, as expected, the net longwave component increases and the net shortwave counterpart decreases. Despite the decreases of the total enthalpy fluxes in most humid regions, the decreases of planetary boundary layer height (PBLH) worldwide (Supplementary Figure S16g) in humid heat days with light rain compared to those without rain can enhance WBGT as well.

Fig. R1. Differences in surface net shortwave (a), net longwave (b), net total radiation (c), latent heat flux (d), sensible heat flux (e), total enthalpy flux (f), and planetary boundary layer height (g) between light rain days and non-rainy days (light rain days minus non-rainy days) in humid heat waves in the STOC simulation.

2) *It's useful to repeat the analyses above for humid heat without rain, with light rain, and with moderate or heavy rain. By analyzing the full surface turbulent and radiative fluxes, and boundary layer depth across three types of humid heat, the authors can more effectively establish that light rain is on a sweet spot for exacerbating humid heat. The authors argue that moderate or heavy rain reduces incoming solar radiation too much and thus diminishes humid heat. Although Figure S21 compares solar radiation between light rain and moderate-to-heavy rain days, how about longwave radiation and surface enthalpy flux? Is the solar radiation reduction in moderate-to-heavy rain days strong enough to cause significant reductions in enthalpy flux? Is it possible that the evapotranspiration becomes energy limited in moderate-to-heavy rain days, so that more water replenishment won't further raise evaporation (or change evaporative fraction)?*

Reply: We thank the reviewer for the valuable suggestions. Fig. R2 further shows the differences between moderate-to-heavy rain and light rain days in humid heat waves. Compared to the light-rain case, surface net shortwave radiation for the moderate-to-heavy rain case is decreased substantially while the longwave counterpart is slightly increased, resulting in a significant decrease in surface net total radiation. Different from the difference between the light-rain case and the no-rain case, the latent heat fluxes changing from the light-rain case to the moderate-to-heavy rain case decrease worldwide. This indicates that more water replenishment will not further increase evapotranspiration which is suppressed by radiation. Along with the decreases of both latent and sensible heat fluxes, there are widespread substantial reductions of the total enthalpy fluxes even over the arid and semi-arid regions, consistent with the changes of surface net total radiation. As a result, compared to the light-rain case, moderate-to-heavy rain is not favorable for humid heat waves although the PBLH is further decreased.

In the revision, the original Fig. S21 is replaced by Fig. R2 (Fig. S17 in the revision), and more clarifications are made in Lines 341-344.

Although moderate to heavy rainfall can offer more surface water than light rain and the associated PBLH is further decreased, the substantial decrease of downward radiation during rainfall effectively suppresses evaporation and thus the total enthalpy flux worldwide, terminating humid heat waves (Supplementary Figure S17).

Fig. R2. Differences in surface net shortwave (a), net longwave (b), net total radiation (c), latent heat flux (d), sensible heat flux (e), enthalpy flux (f), and planetary boundary layer height (g) between moderate-to-heavy rain days and light rain days (moderate-to-heavy rain days minus light rain days) in humid heat waves in the STOC simulation.

Minor concerns

(1) I appreciate the “mathematical description of the impact of light rain on T_w ” in the methods. However, the derivation is problematic. In deriving Eq. 6, the authors assume that both dT_a/dq_s and dq_a/dq_s are zero, which is wrong. We should expect $dT_a/dq_s < 0$ and $dq_a/dq_s > 0$.

When R_s , R_l and conductance parameters in Eq. (4) are held constant, the correct interpretation is that the sum of latent and sensible flux remain invariant to q_s changes. Namely, the moist enthalpy gradient between the surface and the air should be constant in order to maintain a constant enthalpy flux. But we cannot know whether moist enthalpy at the surface or the air will change or not. That is to say, surface energy balance alone does not offer enough constraint. Physically, moist enthalpy in the air could still be altered by other processes like advection, boundary layer depth response,

mixed-layer top entrainment, etc. Land-atmosphere coupling makes sure surface and air moist enthalpy always change at the same rate.

If one assumes all these other processes are not affected by dq_s , then it's reasonable to believe that moist enthalpy at both surface and the air won't change. But, this is not the case in the real world. For example, boundary layer growth will be suppressed under wet soil with less buoyancy flux. This will also change entrainment flux at mixed-layer top.

Reply: We thank the reviewer for noting the issues with the assumptions in the mathematical description. Since we, following the reviewer's major comments, have shown the related processes in the real world, we decide to remove the mathematical description in the revision.

(2) The authors argue that light rain not only makes humid heat longer but also more intense. However, this seems not to be supported by the results. In figure S6, humid heat intensity is slightly weaker with light rain across the majority of the world. In Fig. 3c and Fig. S14a-b, WBGT intensity without rain is also higher than that with rain disregarding the Gini index. So, it seems that light rain will prolong humid heat but has little impact on or even tend to reduce humid heat intensity.

Reply: We thank the reviewer for pointing out this. In the original Fig. S6 (current Fig. S6a), WBGT intensity in arid areas is stronger, which is consistent with the increases in the total enthalpy fluxes (Fig. R1f). In Fig. 3c and d in the revision, in the bin with the largest Gini index, WBGT intensity is slightly larger than or comparable with that without rain. However, the intensification of WBGT becomes more obvious over most regions of the world during consecutive humid heat waves (Fig. S9a in the revision).

In the revised manuscript, we removed the related overstatements such as "intensity of humid heat extremes are significantly exacerbated by light rain", and pointed out that humid heat waves are intensified mainly in arid regions in Lines 64-65, and light rain more likely tend to prolong humid heat compared to the intensity in Lines 152-154.

...humid heat waves can be intensified by light rain mainly in arid and semi-arid regions.

For consecutive events with light rain, light rain prolongs humid heat more efficiently than enhancing the intensity (Supplementary Figure S9).

(3) By showing the Gini index, the authors demonstrate that when a humid heat wave is prolonged by light rain, it's sporadic rain among rain-free days rather than raining every day. This is helpful! But the authors should avoid making statements like "a higher Gini index will make a humid heat wave longer or stronger". That's because the definition of Gini index inherently depends on the heatwave duration. Imagine a precipitation event that lasts two days, the Gini index will be lower if it happens within a 3-day heatwave and higher if it is embedded within a 10-day heatwave. The positive correlation between Gini index and duration is more likely to reflect such inherent

dependence of Gini index calculation on duration, rather than suggesting a higher Gini index makes heat waves longer. My point is it's useful to describe the phenomena, i.e. the uneven distribution of rain, or make hypotheses based on that. But they should avoid going too far making causality arguments that cannot be supported by the results.

Reply: Thank the reviewer for the valuable comment. Following the suggestion, the related statements are rephrased in Lines 165-167, 202-203, 214-215 and 225-226 in revision:

Overall, regions with more uneven distributions of light rain have longer durations of consecutive events (Fig. 2a and b).

As the distribution of rain becomes more uneven, the composited consecutive events, on average, exhibit higher intensity and longer duration (Fig. 3c-f).

As the distribution of rain becomes more uneven, the correlation between latent heat flux and downward solar radiation decreases.

As the distribution of rain becomes more uneven, the rate of latent heat flux change slows down (Fig. 3e and f), consistent with the changes in duration.

(4) In Line 191, the authors argue that “for given intensities and durations, the higher the Gini index, the higher the frequency of consecutive events” which is not the case in Fig. 3. The event frequency first increases with the Gini index and then decreases. It simply reflects the shape of the distribution of Gini index: lower in both tails and higher in-between. It's most obvious in terms of the group of duration. For longer duration, the Gini index frequency peaks at a higher index value, and vice versa. Given this interpretation, I'm not sure how Fig. 3 a-b is useful.

Reply: We thank the reviewer for the valuable comment. We agree that the shape of the distribution of the Gini index can contaminate the analysis of the impact of the uneven distribution of light rain on extreme humid heat. In comparison with the original Fig. 3a and b, since the sum of the columns with light rain and without rain in each row is 100%, the distinct distributions of the Gini index in each row between the two panels imply that the changes of the distribution of the Gini index differ in intensities and durations of consecutive events. We can see that the frequency distribution of the Gini index for all consecutive events with light rain shows a peak at 0.7-0.8 (black line in Fig. R3), as the reviewer noted, with low distributions in both tails and high distributions in between. The distribution of the Gini index for consecutive events with light rain of different intensities shows that as the WBGT intensity increases, the distribution of the Gini index, despite still peaking at 0.7-0.8, becomes more concentrated featuring thinner and taller shapes as WBGT intensifies (Fig. R3a). Different from the intensity, the peaks of the Gini index distribution for different durations shift rightward as the number of days increased (Fig. R3b). Regardless, more

intense and longer consecutive events occur more frequently under more uneven distribution of light rain, and less frequently under less uneven light rain distribution.

We added the related discussion in Lines 191-201 in the revised manuscript, and the original Fig. 3a and b is replaced by Fig. R3.

All consecutive events are grouped by their intensities and durations, though which the occurrence frequency of cases with different Gini indices in light rain events are presented (Fig. 3a and b). The frequency distribution of the Gini index for all consecutive events with light rain shows a peak at Gini index values of 0.7-0.8, with low frequencies in both tails and high frequencies in between. The distribution of the Gini index for consecutive events of different intensities having light rain shows that as the WBGT intensity increases, the distribution of the Gini index, although still peaking at 0.7-0.8, becomes narrower and more spiky. On both sides of the peak the distribution shifts rightward. Different from the intensity, the entire distribution, including the peaks, of the Gini index distribution for different durations, shift rightward as the number of days increases (Fig. 3b). Regardless, more intense and longer consecutive events occur more frequently under more uneven distribution of light rain, and less frequently under less uneven light rain distribution.

Fig.R3. The frequency distribution of Gini index for all consecutive humid heat events with light rain (black line), and separating it into those for different intensities (a) and durations (b) of consecutive humid heat events.

(5) Why is it useful to show that the inter-annual variation of total humid heat days and the days with light rain match well in figure S5? It's not surprising that a major proportion of humid heat days vary consistently with the total humid heat days. How is this information useful?

Reply: We agree with the reviewer. In the revision, the original Fig. S5c-f is removed.

(6) I suggest removing Eq. 1-2 and referring readers to Liljegren et al. (2008) for the details. Eq. 1-2 is rearranged from the energy balance of the WBGT sensor. But this rearrangement also makes it difficult to interpret intuitively (the authors didn't do it

either). So, adding Eq. 1-2 won't help readers understand but introducing lots of unnecessary notations.

Reply: We thank the reviewer for the helpful suggestion. It has been done in Lines 526-528 in the revision:

The Liljegren model is used to calculate T_w and T_g , which are obtained through iterative solutions using temperature, relative humidity, radiation, and wind speed (Eqs. 6 and 9 in Liljegren et al., 2008).

(7) Line 527: change *implement* to *implemented*.

Reply: Done (Lines 531-532 in the revision).

Kong (Kong and Huber 2022) implemented the Liljegren model in Python for the calculation of T_w and T_g .

Reference:

Kong, Q., and M. Huber, 2022: Explicit Calculations of Wet-Bulb Globe Temperature Compared With Approximations and Why It Matters for Labor Productivity. *Earth's Future*, 10, <https://doi.org/10.1029/2021EF002334>.

(8) In Figure 3d, and Figure S14c-d, should the right y-axis be negative?

Reply: The right y-axis, the rate of change of latent heat flux, is the absolute value of the slope of linear regression of latent heat with time. It is because we just focus on the speed of change to characterize the stability of latent heat flux. The larger the absolute value, the faster the change, and vice versa. The slopes can be either positive or negative (e.g., Fig. S11 in the revision). If not done in this way, the positive and negative slopes will cancel out each other when averaged over different Gini index bins (Fig. 3e and f in the revision), which cannot reflect whether the latent heat is maintained at a stable level by light rain. In the revision, more clarifications are added in Lines 219-223:

The rate of latent heat flux change over time, defined as the absolute value of the slope of linear regression of latent heat flux with time, reflects the stability of the change of latent heat flux during an event. The larger the change rate, the faster and more unstable the latent heat flux changes over time, and vice versa. Note that the change rate does not represent whether the latent heat flux becomes larger or smaller.

(9) In Figure S2 and S18, multiplying the 0.7, 0.2 and 0.1 weights on T_w , T_g and T_a makes it hard to distinguish the color scale. Better to show the original value.

Reply: Figs. R4 and R5 show the original values of T_w , T_g and T_a . Fig R4 is added to Fig. S2a, d and g in the revision, and Fig. S18 in the original manuscript is replaced by Fig. R5 (Fig. S13 in the revised manuscript).

Fig. R4. Intensity of natural wet bulb temperature (T_w) (a), black globe temperature (T_g) (b), and dry bulb temperature (T_a) (c).

Fig. R5. Difference in T_w between CAM5 simulation and ERA5 (a) and difference between STOC and CAM5 (b). c, d, Difference in T_g between CAM5 simulation and

ERA5 (c) and the difference between STOC and CAM5 (d). e, f, Difference in T_a between CAM5 simulation and ERA5 (e) and the difference between STOC and CAM5 (f).

(10) Line 51: move citation 20 to the end of the sentence.

Reply: Done (Lines 51-52 in the revision).

Compared to extreme dry heat events, humid heat waves will be more frequent, stronger, and last longer in the future (Wang et al. 2021).

Reference:

Wang, P., Y. Yang, J. Tang, L. R. Leung, and H. Liao, 2021: Intensified Humid Heat Events Under Global Warming. *Geophysical Research Letters*, 48, <https://doi.org/10.1029/2020GL091462>.

(11) There are confusions on the methodological details at several places. Below are some examples:

In figure 1c, is the correlation calculated for summer average WBGT and light rain frequency across the 20-year study period? Similar confusion exists for other places when correlation is calculated.

Reply: In Fig. 1c, the correlation between the light rain frequency and WBGT intensity during the four hottest months is calculated at the monthly scale over the 20-year period. In Fig. 2c, the correlation calculation is applied to the number of days of consecutive events and the associated Gini index over the 20-year period. Same as Fig. 2c, but Fig. 4d and e is for model simulations. In the revision, more clarifications are added in Lines 135-137:

Fig. 1 | Light rain, WBGT and their relationship during the hottest four months. a, b, c, Global distributions of light rain probability (a), WBGT intensity (b) and temporal correlation coefficients between the two at the monthly scale over 2001-2020 (c).

Line 102: How is the 95th percentile calculated? Is it calculated across the whole summer or conditional on some intervals around the target calendar day?

Reply: We calculated the 95th percentile over all the summer days during the reference period of 2001–2020. In Lines 102-103 in the revision, we clarified this:

... humid heat extremes (defined as those days with WBGT intensities exceeding the 95th percentile over the reference period of 2001-2020).

In line 109, the authors state that the correlation is calculated between “the number of days with light rain” and “the number of days with extreme humid heat”. However, the caption of figure 1f seems to state that the former quantity should be “the number of

humid heat days or the day before” with light rain. Please clarify and check other places to avoid similar confusion.

Reply: We clarified this in Lines 111-113 in the revision:

The inter-annual variation of the light rain days during the humid heat days and the day before is positively correlated with that of the number of extreme humid heat days (Fig. 1f).

In figure 1f, why calculate the standard deviations of humid heat days with or without light rain? How to interpret the ratio between these two standard deviations? What does it mean by the “ratio exceeds 0.5 in most regions and approaches 1 in the Amazon”? These seem confusing to me.

Reply: Fig. 1f along with Fig. S5 in the revision aims to see the extent to which the frequency of humid heat waves can be augmented by an increase in light rain days. This is analyzed at the inter-annual scale. By analyzing the ratio of standard deviation between the two, it can be seen that the contribution of the inter-annual variation of the number of light rain days on humid heat wave days and the day before to the inter-annual variation of the total number of humid heat days. The ratio exceeding 0.5 indicates that if the number of humid heat days increases, more than 50% is contributed by the increase of light rain days. More discussion is added in Lines 110-111 and Lines 117-119 in the revision:

To investigate the extent to which the frequency of humid heat waves can be augmented by an increase in light rain days, their interannual variabilities are analyzed.

These ratios indicate that 50-100% of the number of extreme humid heat days increases is contributed by the increase of light rain days.

REVIEWERS' COMMENTS

Reviewer #2 (Remarks to the Author):

I appreciate the authors' efforts in addressing my comments. The manuscript is clearly improved and is ready for publication!